# ReCogDrive: A Reinforced Cognitive Framework for End-to-End Autonomous Driving

Yongkang Li[1,2][*], Kaixin Xiong[2][*], Xiangyu Guo[1,2], Fang Li[2], Sixu Yan[1], Gangwei Xu[1,2],
Lijun Zhou[2], Long Chen[2], Haiyang Sun[2][†], Bing Wang[2], Kun Ma[2], Guang Chen[2],
Hangjun Ye[2], Wenyu Liu[1], Xinggang Wang[1][✉]

[1]Huazhong University of Science and Technology      [2]Xiaomi EV
*{liyk, xgwang}@hust.edu.cn, {sunhaiyang1}@xiaomi.com
**Model & Data:** ReCogDrive HuggingFace Collection
**Code:** ReCogDrive Github Repository

## ABSTRACT

Recent studies have explored leveraging the world knowledge and cognitive capabilities of Vision-Language Models (VLMs) to address the long-tail problem in end-to-end autonomous driving. However, existing methods typically formulate trajectory planning as a language modeling task, where physical actions are output in the language space, potentially leading to issues such as format-violating outputs, infeasible actions, and slow inference speeds. In this paper, we propose ReCogDrive, a novel **Re**inforced **Cog**nitive framework for end-to-end autonomous **Driv**ing, unifying driving understanding and planning by integrating an autoregressive model with a diffusion planner. First, to instill human driving cognition into the VLM, we introduce a hierarchical data pipeline that mimics the sequential cognitive process of human drivers through three stages: generation, refinement, and quality control. Building on this cognitive foundation, we then address the language-action mismatch by injecting the VLM's learned driving priors into a diffusion planner to efficiently generate continuous and stable trajectories. Furthermore, to enhance driving safety and reduce collisions, we introduce a Diffusion Group Relative Policy Optimization (DiffGRPO) stage, reinforcing the planner for enhanced safety and comfort. Extensive experiments on the NAVSIM and Bench2Drive benchmarks demonstrate that ReCogDrive achieves state-of-the-art performance. Additionally, qualitative results across diverse driving scenarios and DriveBench highlight the model's scene comprehension.

## 1 INTRODUCTION

Autonomous driving, which aims to predict a smooth, comfortable, and collision-free trajectory for a vehicle, has seen significant advancements. Recent end-to-end autonomous driving systems (Jiang et al., 2023; Hu et al., 2023; Chen et al., 2024b; Hu et al., 2022; Zhang et al., 2024a) unify the perception (Jiang et al., 2024b; Philion & Fidler, 2020; Zhang et al., 2023), prediction (Chai et al., 2019; Gu et al., 2023), and planning (Chitta et al., 2022; Liao et al., 2024) modules into a single pipeline for joint optimization, demonstrating impressive performance under open-loop evaluation. However, these systems often fail to generalize to long-tail scenarios, where data is limited and the driving conditions deviate significantly from those encountered during training.

Recent research (Tian et al., 2024; Jiang et al., 2024a; Hwang et al., 2024) addresses the long-tail challenge by introducing VLMs, which are pre-trained on large-scale internet datasets and exhibit strong generalization abilities along with rich world knowledge. Specifically, VLMs applications in autonomous driving can be categorized into two main approaches: (1) *dual-system* approaches (Jiang et al., 2024a; Tian et al., 2024), where VLMs generate low-frequency trajectories or high-level commands to guide an end-to-end driving system; and (2) *single-system* approaches (Hwang et al.,

---

*This work was done when Yongkang Li was an intern at Xiaomi EV. * Equal contribution. † Project leader.
✉ Corresponding author: xgwang@hust.edu.cn.

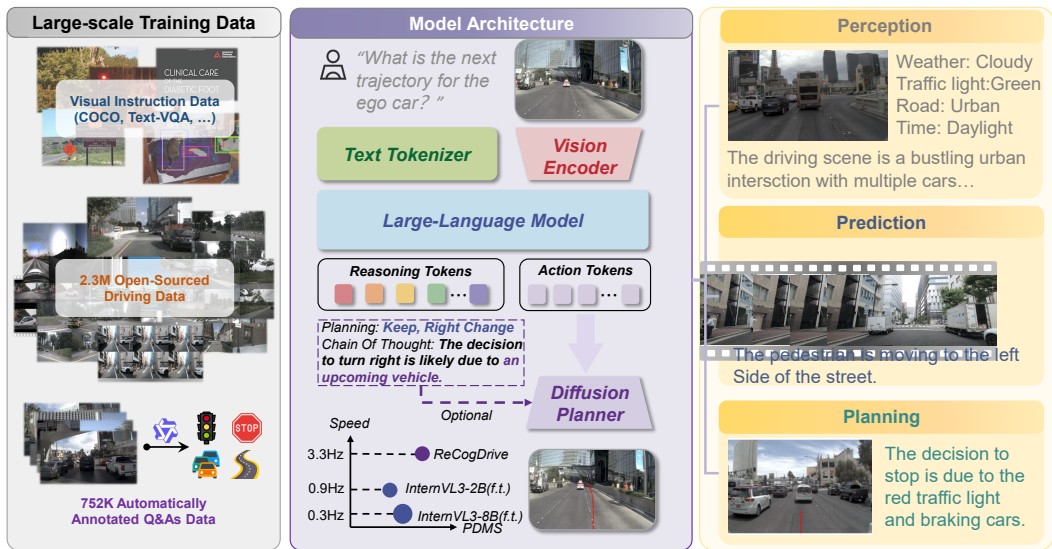

Figure 1: **Overview of ReCogDrive.** We present ReCogDrive, an end-to-end autonomous driving system, which possesses rich driving priors and generates continuous, stable trajectories via a diffusion denoising process. ReCogDrive is capable of performing tasks spanning from low-level scene perception and motion prediction to high-level driving planning and decision making.

2024; Mao et al., 2023a; Xing et al., 2024; Mao et al., 2023b; Wang et al., 2024b; Bai et al., 2024; Zhang et al., 2024c; Zhao et al., 2025), where standalone VLMs directly predict future trajectories and can be optimized in an end-to-end manner. Most of these approaches reformulate the motion planning task as a language modeling problem, generating trajectories in a textual format through autoregressive prediction. These methods leverage the generalization of the models and chain-of-thought (Wei et al., 2022) reasoning to enhance interpretability and reasoning in complex scenarios.

Despite these advantages, directly applying pre-trained VLMs by reformulating trajectory planning as a text generation task for autonomous driving reveals three critical limitations: (1) *Domain discrepancy of pre-trained knowledge.* VLMs (Mao et al., 2023a;b) are trained on generic internet data that lacks the nuanced knowledge required for driving, resulting in a significant domain gap. (2) *Modality mismatch in trajectory generation.* The discrete language space of VLMs fundamentally conflicts with the continuous action space required for planning. This mismatch leads to practical issues, as the probabilistic nature of autoregressive decoding can lead not only to physically infeasible trajectories but also to format-violating outputs that result in parsing errors. (3) *Suboptimal policies from imitation learning.* The heavy reliance on behavior cloning causes models (Hwang et al., 2024) to converge to a suboptimal policy that struggles in rare scenarios and can be unsafe.

To address these challenges, we propose ReCogDrive, a novel end-to-end autonomous driving system that integrates the cognitive reasoning of Vision-Language Models (VLMs) with a reinforcement learning-enhanced diffusion planner. Our framework introduces three key innovations: First, to bridge the domain gap, we design a hierarchical data pipeline that, through three stages of generation, refinement, and quality control, constructs a large-scale VQA dataset to instill human-like cognitive priors into the VLM. Second, to resolve the modality mismatch, we design a diffusion planner that effectively translates the VLM's latent cognitive representations into continuous and stable driving trajectories. Finally, to overcome the limitations of imitation learning, we introduce a reinforcement learning stage based on Diffusion Group Relative Policy Optimization (DiffGRPO), enabling the planner to explore safer, more optimal behaviors beyond the expert dataset.

We extensively evaluate ReCogDrive on NAVSIM and Bench2Drive, achieving new state-of-the-art performance in both open-loop and closed-loop settings. Additionally, we test ReCogDrive-VLM on the DriveLM and DriveBench benchmarks, demonstrating its superiority over both closed-source generalist models and open-source expert models. Qualitative analyses further demonstrate that our reinforcement learning method enables the model to drive more safely and comfortably while significantly reducing collisions. The main contributions of this work are as follows:

- We propose ReCogDrive, a novel end-to-end autonomous driving framework that integrates a cognitive VLM with a diffusion planner, leveraging the rich world knowledge of VLMs to provide cognitive guidance and enabling stable and precise trajectory generation.

- We propose Diffusion Group Relative Policy Optimization (DiffGRPO) to fine-tune the planner, allowing the model to learn directly from experience and optimize for safety and comfort, moving beyond mere imitation.

- We establish new state-of-the-art performance on NAVSIM and Bench2Drive in both open-loop and closed-loop evaluations, demonstrating significant improvements over prior approaches.

## 2 RELATED WORK

**Vision-Language-Models in Autonomous Driving.** Numerous studies have leveraged the world knowledge embedded in VLMs to explore their application in autonomous driving. Current approaches for autonomous driving using VLMs can be categorized primarily into two types: *dual-system* (Tian et al., 2024; Jiang et al., 2024a) and *single-system* (Hwang et al., 2024; Mao et al., 2023a; Xing et al., 2024; Mao et al., 2023b; Wang et al., 2024b; Bai et al., 2024; Shao et al., 2024a; Zhang et al., 2024c; Zhao et al., 2025; Fu et al., 2025; Zhou et al., 2025a). Dual-system methods, such as DriveVLM and Senna, integrate VLMs with end-to-end driving systems, where VLMs generate low-frequency trajectories or high-level commands that guide the end-to-end model in producing the final trajectory. In addition, several recent approaches (Pan et al., 2024; Hegde et al., 2025) distill VLM knowledge into end-to-end planners, further improving generalization and robustness. In contrast, single-system methods, such as GPT-Driver and EMMA, reformulate the trajectory prediction task as a language modeling problem and leverage chain-of-thought (Wei et al., 2022) to enhance explainability. However, these text-based approaches often suffer from high inference latency and malformed outputs, whereas our ReCogDrive framework combines a VLM with a diffusion planner to produce more stable, safer trajectories while achieving a 3.5× speedup.

**Diffusion Models for Policy Learning.** The recent success of diffusion models in domains like image generation (Podell et al., 2023; Zheng et al., 2024), robotics (Chi et al., 2023; Liu et al., 2024b; Black et al., 2024; Yan et al., 2025; Bjorck et al., 2025), and traffic simulation (Yang et al., 2024; Zhong et al., 2023; Zeng et al., 2025) has spurred their application in autonomous driving. For instance, DiffusionDrive (Liao et al., 2024) employs truncated diffusion with anchor priors for real-time multimodal planning. GoalFlow (Xing et al., 2025) applies goal-point guidance with flow matching to generate high-quality trajectories. Diffusion Planner (Zheng et al., 2025) redefines planning as a future trajectory generation task, jointly producing plans and motion forecasts. Our approach uses world knowledge from VLMs to guide the diffusion process, thereby improving the model's understanding of driving scenarios.

**Reinforcement Learning in Autonomous Driving.** Reinforcement Learning has proven effective in games (Hafner et al., 2023), robotics (Ren et al., 2024), and LLMs (Guo et al., 2025). Recently, many studies have introduced reinforcement learning into autonomous driving to improve model generalization. RAD (Gao et al., 2025) proposes training an end-to-end AD agent using Reinforcement Learning in a photorealistic 3DGS environment. CarPlanner (Zhang et al., 2025) learns an auto-regressive policy for consistent multi-modal trajectories, outperforming imitation learning. AlphaDrive (Jiang et al., 2025), R2SE (Liu et al., 2025), Drive-R1 Li et al. (2025c) and TrajHF (Li et al., 2025a) incorporate GRPO to enhance the generalization of driving policies. Gen-Drive (Huang et al., 2025) combines diffusion models with RL and reward modeling for better policies. Our method novelly employs a Diffusion Group Relative Policy Optimization (DiffGRPO) scheme to fine-tune Vision-Language-Action (VLA) models and enhance their planning capabilities.

## 3 PRELIMINARIES

**Problem Definition.** Autonomous driving aims to predict smooth and collision-free trajectory in future seconds, given the ego status $S_{\text{ego}}$ (e.g., ego speed $\in \mathbb{R}^2$ containing $v_x, v_y$ and ego acceleration $\in \mathbb{R}^2$ containing $a_x, a_y$), the historical ego trajectory $I_{\text{hist}} \in \mathbb{R}^{T_{\text{hist}} \times 3}$ (containing past waypoints and headings), sensor input $I_{\text{cam}} \in \mathbb{R}^{K \times V \times H \times W \times 3}$ (containing current and historical $K$ frames from $V$ camera views), and navigation information $L_{\text{nav}}$ (High-level command such as "go straight").

Conventional end-to-end driving algorithm $\Phi$ formulate this as

$$\mathbf{V}_{\text{traj}} = \Phi\big(I_{\text{cam}},\, I_{\text{hist}},\, L_{\text{nav}},\, S_{\text{ego}}\big), \tag{1}$$

where $\mathbf{V}_{\text{traj}}$ denotes the planned future trajectory, parameterized as $\{(x_t, y_t, \theta_t)\}_{t=1}^{T} \in \mathbb{R}^{T \times 3}$, with $(x_t, y_t)$ the 2D waypoints and $\theta_t$ the corresponding heading angles. Explicit headings are needed to construct oriented ego polygons for collision evaluation and to derive yaw-related comfort measures, which cannot be robustly obtained from sparsely sampled waypoints alone. While methods such as (Hu et al., 2023; Jiang et al., 2023; Liao et al., 2024; Chen et al., 2024b) have shown strong effectiveness, their black-box nature impedes interpretability and they often fail to generalize to rare corner cases in real-world driving.

Recent works (Hwang et al., 2024; Mao et al., 2023a; Zhang et al., 2024b) utilize the rich world knowledge and causal reasoning capabilities of Vision–Language Models for autonomous driving. VLMs output trajectories in textual form and generate explicit reasoning processes:

$$T_{\text{traj}},\, T_{\text{reason}} = \text{VLM}\big(I_{\text{cam}},\, I_{\text{hist}},\, L_{\text{nav}},\, S_{\text{ego}}\big). \tag{2}$$

However, we observe an inherent mismatch between the language-formatted trajectory space and the continuous action space, and the autoregressive decoding process can suffer from output collapse, leading to erroneous trajectories.

**Diffusion Policy.** Denoising Diffusion Probabilistic Models (DDPMs) (Ho et al., 2020; Nichol & Dhariwal, 2021) learn a generative model by reversing a fixed, Markovian noising process that gradually corrupts data with Gaussian noise. Given a clean trajectory $\mathbf{x}_0$, the forward process defines:

$$q(\mathbf{x}_t \mid \mathbf{x}_{t-1}) = \mathcal{N}\big(\mathbf{x}_t;\, \sqrt{1 - \sigma_t^2}\,\mathbf{x}_{t-1},\, \sigma_t^2 \mathbf{I}\big), \quad t = 1, \ldots, T, \tag{3}$$

where $\sigma_t$ is the noise standard deviation at step $t$. At inference, trajectories are generated by initializing $\mathbf{x}_T \sim \mathcal{N}(\mathbf{0}, \mathbf{I})$ and iteratively denoising:

$$\mathbf{x}_{t-1} = \frac{1}{\sqrt{1 - \sigma_t^2}}\Big(\mathbf{x}_t - \sigma_t^2\, \epsilon_\theta(\mathbf{x}_t, t)\Big) + \sigma_t\, \mathbf{z}, \quad \mathbf{z} \sim \mathcal{N}(0, \mathbf{I}). \tag{4}$$

Denoting trajectory waypoint as $\mathbf{x}_0$, this framework naturally extends to trajectory-level policy generation, where the denoising network $\epsilon_\theta$ learns to refine noisy motion plans into smooth trajectories.

## 4 METHODOLOGY

In this section, we present the overall framework and training paradigm of ReCogDrive. First, we introduce a scalable hierarchical data pipeline that curates a multi-level cognitive dataset spanning from perception to reasoning to instill the VLM with nuanced, human-like driving priors. Second, we detail our cognition-guided diffusion planner, a novel architecture that resolves the modality mismatch by translating the VLM's latent semantic representations into continuous and stable trajectories. Finally, we describe our policy refinement stage, where a Diffusion Group Relative Policy Optimization (DiffGRPO) algorithm is employed to fine-tune the planner with simulated rewards, optimizing for safety and comfort beyond the confines of the expert dataset.

### 4.1 SCALABLE HIERARCHICAL DATA PIPELINE FOR DRIVING PRETRAINING

To bridge the gap between general Vision-Language Models (VLMs) and autonomous driving, we introduce a scalable, structured pipeline for the automated generation of high-fidelity driving data, as shown in Fig. 2. This pipeline comprises three stages: Generation, Refinement, and Quality Control.

In the Generation stage, we align with the sequential cognitive process of human drivers, organized into four levels: *(1) Foundational Perception:* constructing representations of static and dynamic scene elements; *(2) Dynamic Understanding:* analyzing multi-agent dynamics and predicting near-term behaviors of surrounding entities; *(3) Planning and Reasoning:* formulating executable, safety-aware driving plans and providing concise rationales; and *(4) Advanced Reasoning:* conducting counterfactual analyses and fine-grained trade-off evaluations. To achieve this, we combine ground-truth data for objective tasks with a state-of-the-art VLM for subjective annotations.

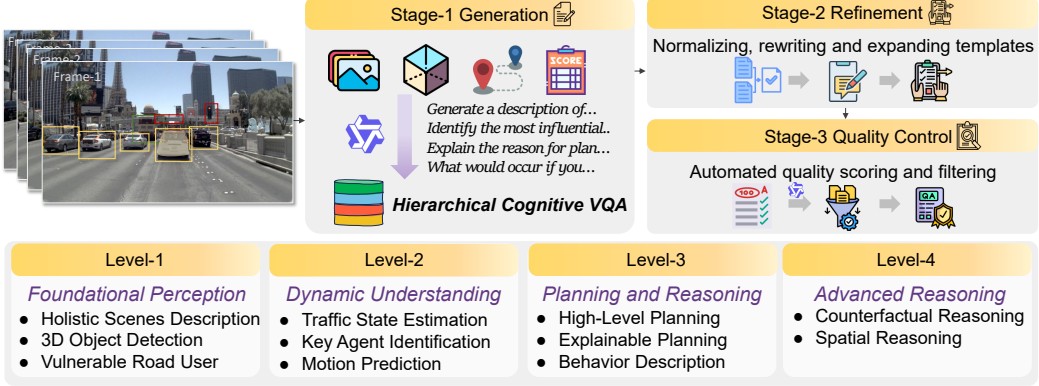

Figure 2: **Overview of our Scalable Hierarchical Data Pipeline.** We employ a three-stage process of Generation, Refinement, and Quality Control to produce a high-quality dataset that mimics the cognitive process of a human driver.

The Refinement stage integrates open-source datasets, followed by normalization, rewriting, and question-template augmentation to ensure linguistic and semantic consistency. Finally, the Quality Control stage applies automated scoring and rigorous filtering based on linguistic accuracy, visual clarity, etc., retaining only the high quality samples. This systematic design ensures our pre-training dataset is both reliable and scalable, enabling the adaptation of a general VLM into a domain-specialized cognitive model for autonomous driving.

## 4.2 COGNITION-GUIDED DIFFUSION PLANNER

While autoregressive paradigms offer a straightforward way to generate trajectories in textual form, they suffer from inherent limitations: (i) the mismatch between discrete language space and continuous action space often leads to infeasible or jerky trajectories, and (ii) VLMs are prone to hallucination and decoding errors, which compromise safety in autonomous driving. To address these issues, we propose a *Cognition-Guided Diffusion Planner*, which integrates human cognitive driving priors from a VLM with a diffusion planner to generate smooth and safety-aware trajectories. Formally, given a noisy trajectory sample $\mathbf{x}_t \in \mathbb{R}^{N \times 3}$, the denoising step is defined as:

$$\mathbf{x}_{t-1} = D_{\text{act}}\Big(\text{DiT}_\theta(z_t;\, F_h;\, S_{\text{ego}};\, t)\Big), \quad z_t = \text{concat}\big(E_{\text{act}}(\mathbf{x}_t),\, E_{\text{his}}(I_{\text{hist}}),\, \bar{F}_h\big), \quad (5)$$

where $F_h$ denotes the VLM hidden states, $S_{\text{ego}}$ represents the ego-vehicle status, and $z_t$ is the fused latent combining the noisy-action latent from the action encoder $E_{\text{act}}$, the historical-trajectory latent of $I_{\text{hist}}$ encoded by the history-trajectory encoder $E_{\text{his}}$, and the semantic prior $\bar{F}_h$ obtained by mean-pooling over $F_h$. And $D_{act}$ is the action decoder that predicts the next step noisy trajectory. The diffusion model is trained by minimizing:

$$\mathcal{L}_{\text{dif}} = \mathbb{E}_{z_t,c} \left\| \epsilon - \epsilon_\pi(z_t, c) \right\|^2, \quad (6)$$

with $\epsilon \sim \mathcal{N}(0, \mathbf{I})$ and condition $c = \{F_h, S_{\text{ego}}\}$.

**Vision-Language Model Guidance.** We adopt InternVL3 (Zhu et al., 2025; Chen et al., 2024e) as the cognitive backbone for its strong multi-modal reasoning. The VLM encodes multi-view images and textual queries into hidden states $F_h \in \mathbb{R}^{L \times D}$, which act as cognitive tokens carrying driving priors and serve as cross-attention conditions in the diffusion transformer. We further obtain a global semantic embedding $\bar{F}_h$ via average pooling over $F_h$, providing stable contextual guidance during denoising. Additionally, the VLM can optionally output high-level driving instructions and chain-of-thought reasoning to enhance trajectory generation.

**Diffusion Transformer Design.** Our diffusion planner builds on DiT (Peebles & Xie, 2023; Yao et al., 2025) blocks, enhanced with lightweight but effective design choices: SwiGLU FFN (Shazeer, 2020) for expressive non-linearity, RoPE (Su et al., 2024) for relative positional encoding, and both

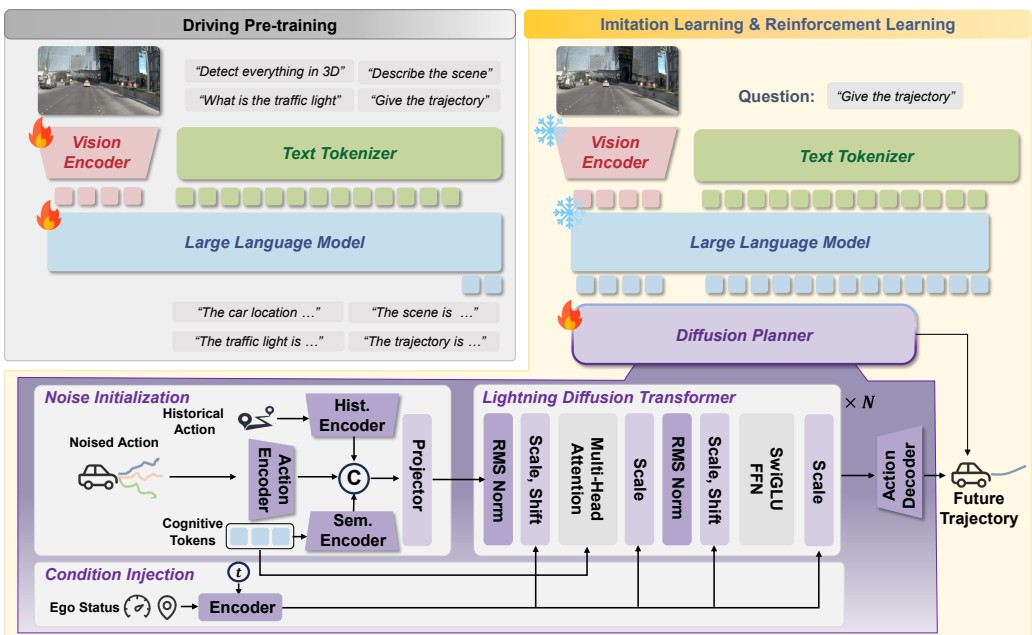

Figure 3: **Training Pipeline and Model Architecture.** ReCogDrive couples a VLM with a diffusion planner: the VLM encodes inputs into cognitive tokens guiding trajectory denoising. Training follows three stages: Driving Pre-training, imitation learning, and reinforcement learning.

QK-Norm and RMSNorm (Zhang & Sennrich, 2019) for stable optimization. Each block alternates between (i) *self-attention*, which models pairwise waypoint relations, and (ii) *cross-attention*, which injects semantic priors $F_h$ into the trajectory space. This design enables a structured fusion of scene-level cognition with low-level trajectory optimization.

**Historical and Ego Information Fusion.** To capture temporal consistency, historical trajectories $\mathbf{x}_{\text{hist}}$ are encoded and concatenated with the noisy trajectory embeddings. Ego-vehicle status $S_{ego}$ (e.g., speed, acceleration) is injected via *AdaLN modulation*, ensuring that the planner is conditioned on the vehicle's physical state. This dual conditioning mechanism enhances stability, safety, and adaptability in diverse driving scenarios.

### 4.3 DIFFUSION GROUP RELATIVE POLICY OPTIMIZATION

Relying solely on imitation learning exhibits fundamental limitations (Chu et al., 2025; Osa et al., 2018; Ren et al., 2024), since expert demonstrations may be multi-modal, leading to conflictual optimization results. As shown in Fig. 4(a), when trained in this rare intersection turn scenario with multiple expert trajectories, the model resorts to learning an average trajectory to achieve global optimality, which can lead to incorrect or unsafe maneuvers. Consequently, even though the diffusion planner trained through imitation learning closely matches expert trajectories in terms of displacement, it still frequently produces collisions or drives outside the drivable area.

To enable exploration beyond imitation, we introduce *DiffGRPO*, a Group Relative Policy Optimization algorithm specifically designed for diffusion planners that tightly couples GRPO (Shao et al., 2024b) with the denoising process. We adopt GRPO since it naturally samples multiple candidate trajectories, which aligns with autonomous driving where multiple feasible ground truth trajectories can satisfy the same scene and high level intent. Following (Ren et al., 2024), we interpret the diffusion policy $\pi_\theta$ as an internal Markov decision process: starting from Gaussian noise, the model gradually denoises to generate a full trajectory. Concretely, we sample $G$ trajectories, each represented by a diffusion chain:

$$\mathbf{x} = (x_T, x_{T-1}, \ldots, x_0),\tag{7}$$

where $T$ is the total number of denoising steps and transitions follow:

$$x_T \sim \mathcal{N}(0, \mathbf{I}), \qquad x_{t-1} \sim \pi_\theta(x_{t-1} \mid x_t).\tag{8}$$

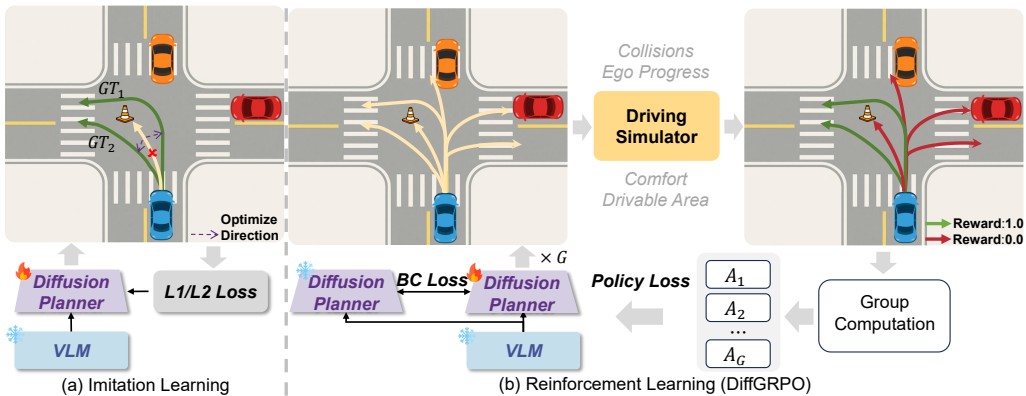

Figure 4: **Comparison of Training Paradigms.** (a) **Imitation Learning:** the diffusion planner is trained offline to mimic ground truth trajectories using L1/L2 losses, but tends to learn averaged, suboptimal paths. (b) **Reinforcement Learning:** multiple trajectories are sampled and evaluated in the NAVSIM simulator, scored on collision avoidance, drivable area compliance and other metrics, and advantages are computed via group computation to update the diffusion planner.

Unlike prior approaches (Li et al., 2025a;c) that rely on simple $\ell_2$ trajectory distances as surrogate rewards, we leverage the NAVSIM simulator to providing realistic feedback on safety and comfort. Each rollout is evaluated in terms of collisions, drivable-area compliance, and driving comfort, with the results aggregated into a Predictive Driver Model Score (PDMS) that serves as the reward $r_i$. We cast this as a single-step decision, treating the whole trajectory as one composite action with its PDMS score as the reward. We then compute the group-standardized advantage:

$$\hat{A}_i = \frac{r_i - \mu}{\sigma}, \quad \mu = \frac{1}{G}\sum_{j=1}^{G} r_j, \quad \sigma^2 = \frac{1}{G}\sum_{j=1}^{G}(r_j - \mu)^2, \quad i = 1, \ldots, G. \tag{9}$$

Each conditional step in the diffusion chain is a Gaussian policy:

$$\pi_\theta\big(x_{t-1} \mid x_t\big) = \mathcal{N}\Big(x_{t-1}; \mu_\theta(x_t, t), \sigma_t^2 I\Big), \tag{10}$$

where $\mu_\theta(x_t, t)$ is the model-predicted mean and $\sigma_t^2 I$ the (fixed) covariance. Thus, the probability density of the full chain $\mathbf{x}_{0:T}$ under $\pi_\theta$, where $p(x_T)$ is a Gaussian prior independent of $\theta$, is:

$$\log \pi_\theta(\mathbf{x}_{0:T}) = \log p(x_T) + \sum_{t=1}^{T} \log \pi_\theta(x_{t-1} \mid x_t), \tag{11}$$

Finally, we compute the policy loss following (Williams, 1992; Shao et al., 2024b) while concurrently incorporating a behavior cloning loss to prevent collapse during exploration.

$$L = \underbrace{-\frac{1}{G}\sum_{i=1}^{G}\frac{1}{T}\sum_{t=1}^{T}\gamma^{t-1}\log \pi_\theta\big(x_{t-1}^{(i)} \mid x_t^{(i)}\big)\hat{A}_i}_{L_{\mathrm{RL}}} - \underbrace{\lambda\frac{1}{G}\sum_{i=1}^{G}\frac{1}{T}\sum_{t=1}^{T}\log \pi_\theta\big(\tilde{x}_{t-1}^{(i)} \mid \tilde{x}_t^{(i)}\big)}_{L_{\mathrm{BC}}}, \tag{12}$$

where $\gamma$ is the discount coefficient mitigating instability in early denoising steps, $\lambda$ is the weight for the behavior cloning loss, and $\tilde{x}_{t-1}, \tilde{x}_t$ are values sampled from the reference policy $\pi_{\mathrm{ref}}$. We omit the PPO-style clipping and set the update iteration to 1, so $\pi = \pi_{\mathrm{old}}$. Through DiffGRPO, the diffusion planner learns to generate safe and comfortable trajectories in closed-loop settings, going beyond mere imitation and enhancing the robustness of our framework.

Table 1: **Performance comparison on NAVSIM *navtest* using closed-loop metrics.** [†] denotes models fine-tuned on the NAVSIM trajectory dataset.

| Method | Image | Lidar | NC↑ | DAC↑ | TTC↑ | Comf. ↑ | EP↑ | PDMS↑ |
|---|---|---|---|---|---|---|---|---|
| Constant Velocity | | | 68.0 | 57.8 | 50.0 | **100** | 19.4 | 20.6 |
| Ego Status MLP | | | 93.0 | 77.3 | 83.6 | **100** | 62.8 | 65.6 |
| VADv2-$\mathcal{V}_{8192}$ (Chen et al., 2024b) | ✓ | | 97.2 | 89.1 | 91.6 | **100** | 76.0 | 80.9 |
| DrivingGPT (Chen et al., 2024c) | ✓ | | **98.9** | 90.7 | 94.9 | 95.6 | 79.7 | 82.4 |
| UniAD (Hu et al., 2023) | ✓ | | 97.8 | 91.9 | 92.9 | **100** | 78.8 | 83.4 |
| TransFuser (Chitta et al., 2022) | ✓ | ✓ | 97.7 | 92.8 | 92.8 | **100** | 79.2 | 84.0 |
| PARA-Drive (Weng et al., 2024) | ✓ | | 97.9 | 92.4 | 93.0 | 99.8 | 79.3 | 84.0 |
| DRAMA (Yuan et al., 2024) | ✓ | ✓ | 98.0 | 93.1 | 94.8 | **100** | 80.1 | 85.5 |
| Hydra-MDP-$\mathcal{V}_{8192}$-W-EP (Li et al., 2024) | ✓ | ✓ | 98.3 | 96.0 | 94.6 | **100** | 78.7 | 86.5 |
| DiffusionDrive (Liao et al., 2024) | ✓ | ✓ | 98.2 | 96.2 | 94.7 | **100** | 82.2 | 88.1 |
| WoTE (Li et al., 2025b) | ✓ | ✓ | 98.5 | 96.8 | **94.9** | 99.9 | 81.9 | 88.3 |
| **VLMs-based Methods** | | | | | | | | |
| QwenVL2.5-8B[†] (Bai et al., 2025) | ✓ | | 97.8 | 92.1 | 92.8 | **100** | 78.3 | 83.3 |
| InternVL3-8B[†] (Zhu et al., 2025) | ✓ | | 97.0 | 92.4 | 91.8 | **100** | 78.9 | 83.3 |
| AutoVLA Zhou et al. (2025b) | ✓ | | 98.4 | 95.6 | **98.0** | 99.9 | 81.9 | 89.1 |
| *ReCogDrive(ours)* | ✓ | | 97.9 | **97.3** | 94.9 | **100** | **87.3** | **90.8** |

Table 2: **Closed-loop and Multi-ability Testing Results in CARLA Bench2Drive Leaderboard.**

| Method | Closed-loop Metric ↑ | | | | Multi-Ability Test (%) ↑ | | | | | |
|---|---|---|---|---|---|---|---|---|---|---|
| | Efficiency | Comfort | Success | **DS** | Merging | Overtaking | Emerg. Brake | GiveWay | Traf. Sign | **Mean** |
| TCP* (Wu et al., 2022) | 54.26 | 47.80 | 15.00 | 40.70 | 16.18 | 20.00 | 20.00 | 10.00 | 6.99 | 14.63 |
| TCP-ctrl* (Wu et al., 2022) | 55.97 | **51.51** | 7.27 | 30.47 | 10.29 | 4.44 | 10.00 | 10.00 | 6.45 | 8.23 |
| TCP-traj* (Wu et al., 2022) | 76.54 | 18.08 | 30.00 | 59.90 | 8.89 | 24.29 | 51.67 | 40.00 | 46.28 | 34.22 |
| TCP-traj w/o distill. (Wu et al., 2022) | 78.78 | 22.96 | 30.05 | 49.30 | 17.14 | 6.67 | 40.00 | 40.00 | 28.72 | 28.51 |
| ThinkTwice (Jia et al., 2023b) | 76.93 | 16.22 | 3.13 | 62.44 | 27.38 | 18.42 | 35.82 | **50.00** | 54.23 | 37.17 |
| DriveAdapter* (Jia et al., 2023a) | 70.22 | 16.01 | 33.08 | 64.22 | 28.82 | 26.38 | 48.76 | **50.00** | 56.43 | **42.08** |
| AD-MLP (Zhai et al., 2023) | 48.45 | 22.63 | 0.00 | 18.05 | 0.00 | 0.00 | 0.00 | 0.00 | 4.35 | 0.87 |
| UniAD-T. (Hu et al., 2023) | 123.92 | 47.04 | 13.18 | 40.73 | 8.89 | 9.33 | 20.00 | 20.00 | 15.43 | 14.73 |
| UniAD-B. (Hu et al., 2023) | 129.21 | 43.58 | 16.36 | 45.81 | 14.10 | 17.78 | 21.67 | 10.00 | 14.21 | 15.55 |
| VAD (Jiang et al., 2023) | **157.94** | 46.01 | 15.00 | 42.35 | 8.11 | 24.44 | 18.64 | 20.00 | 19.15 | 18.07 |
| DriveTransformer-L. (Jia et al., 2025) | 100.64 | 46.01 | 35.01 | 63.46 | 17.57 | **35.00** | 48.36 | 40.00 | 52.10 | 38.60 |
| **ReCogDrive** | 138.18 | 17.45 | **45.45** | **71.36** | **29.73** | 20.00 | **69.09** | 20.00 | **71.34** | 42.03 |

## 5 EXPERIMENTS

### 5.1 EXPERIMENTAL SETUP.

**Implementation Details.** We choose InternVL3 (Zhu et al., 2025), comprising a 300M-parameter InternViT visual encoder (Chen et al., 2024e) and a Qwen2.5 Large-Language-Model (Bai et al., 2025), as our base model, which demonstrates strong performance on multiple benchmarks. Images are processed through a dynamic resolution preprocessing strategy. In the first stage, we conduct supervised fine-tuning (SFT) on the VLM using the dataset constructed by our hierarchical data pipeline for three epochs. In the second stage, with the VLM parameters frozen, we train the diffusion model via DDPM for 200 epochs. In the third stage, we further optimize the diffusion model using reinforcement learning for 10 epochs, with one policy update per batch. The datasets and training hyperparameters for all three stages are described in detail in the supplementary material.

**Dataset.** We evaluate primarily on two challenging benchmarks: NAVSIM (Dauner et al., 2025) and Bench2Drive (Jia et al., 2024). NAVSIM is a planning-oriented autonomous driving dataset built on OpenScene (Contributors, 2023), a redistribution of nuPlan (Caesar et al., 2021). The dataset is split into *navtrain* (1,192 training scenes) and *navtest* (136 evaluation scenes). Bench2Drive is a CARLA-based benchmark composed of 220 short routes, each containing a distinct, safety-critical scenario. In addition, to build the VLM's cognitive foundation, ReCogDrive is trained on a collection of VQA datasets, including DriveLM (Sima et al., 2024) and LingoQA (Marcu et al., 2024), with further details provided in the supplementary material.

Table 3: **Ablation study on the proposed components of ReCogDrive.** We evaluate the effect of driving pre-training, diffusion planner, and reinforcement learning on NAVSIM evaluation.

| ID | Trajectory training | Driving Pre-training | Diffusion Planner | Diff GRPO | NC | DAC | TTC | Conf. | EP | PDMS |
|---|---|---|---|---|---|---|---|---|---|---|
| 1 | ✓ | ✗ | ✗ | ✗ | 97.4 | 91.3 | 93.0 | 100 | 77.2 | 82.4 |
| 2 | ✓ | ✓ | ✗ | ✗ | 97.6 | 93.1 | 92.7 | 100 | 79.1 | 84.1 |
| 3 | ✓ | ✓ | ✓ | ✗ | 98.1 | 94.7 | 94.2 | 100 | 80.9 | 86.5 |
| 4 | ✓ | ✓ | ✓ | ✓ | 97.9 | 97.3 | 94.9 | 100 | 87.3 | 90.8 |

Table 4: Comparison of trajectory generation methods.† indicates inference with `lmdeploy`.

| Method | Time (s) | PDMS ↑ | Err. (%) ↓ |
|---|---|---|---|
| VLM Plain Text† | 1.0702 | 84.1 | 0.01 |
| VLM + MLP | 0.3045 | 75.5 | 0.00 |
| VLM + Query-based Decoder | 0.3050 | 85.0 | 0.00 |
| VLM + Vanilla Diffusion | 0.3060 | 85.4 | 0.00 |
| + *SwiGLU FFN* | 0.3065 | 85.7 | 0.00 |
| + *RMS Norm* | 0.3070 | 85.8 | 0.00 |
| + *RoPE & QK Norm* | 0.3061 | 86.5 | 0.00 |

Table 5: Model performance on Drive VQA Benchmark.

| Method | DriveLM | DriveBench | | | | |
|---|---|---|---|---|---|---|
| | GPT-Score | Percep. | Predict. | Plan. | Behav. | Avg. |
| LLaVA-1.5 | 61.91 | 23.22 | 22.02 | 29.15 | 13.60 | 22.00 |
| InternVL2 | 64.13 | 32.36 | 45.52 | 53.27 | 54.58 | 46.43 |
| Qwen2-VL | – | 30.13 | 49.35 | 61.30 | 51.26 | 48.01 |
| DriveLM | 65.25 | 16.85 | 44.33 | 68.71 | 42.78 | 43.17 |
| Dolphins | – | 9.59 | 32.66 | 52.91 | 8.81 | 25.99 |
| GPT-4o | 67.27 | 35.37 | 51.30 | 75.75 | 45.40 | 51.96 |
| ReCogDrive | **67.30** | 64.95 | 49.34 | 70.20 | 42.36 | **56.71** |

Table 6: **Comparison of different RL algorithms.** We evaluate REINFORCE, DPPO, and DiffGRPO on our diffusion planner under the same training and evaluation setting.

| Algorithm | NC | DAC | TTC | Conf. | EP | PDMS |
|---|---|---|---|---|---|---|
| REINFORCE | 98.5 | 97.3 | 95.9 | 100 | 82.9 | 89.5 |
| DPPO | 98.6 | 97.6 | 96.0 | 100 | 82.4 | 89.5 |
| DiffGRPO | 97.9 | 97.3 | 94.9 | 100 | 87.3 | **90.8** |

Figure 5: **Training curves on NAVSIM.**

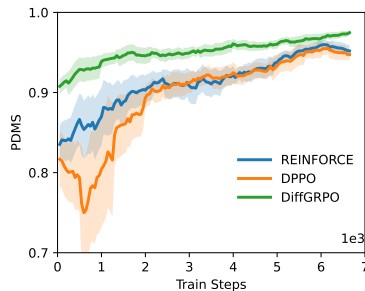

## 5.2 MAIN RESULTS AND ABLATION STUDY

**Experiments on the NAVSIM Benchmark.** Tab. 1 reports results on NAVSIM. ReCogDrive achieves a PDMS of 90.8, establishing a new state-of-the-art. Remarkably, it outperforms Diffusion-Drive (Liao et al., 2024) and WoTE (Li et al., 2025b), both of which use camera and LiDAR inputs, by 2.7 and 2.5 PDMS, respectively, while using only camera input. Compared with our reproduced InternVL3 (Zhu et al., 2025) and QwenVL2.5 (Bai et al., 2025) trained directly on NAVSIM trajectories, ReCogDrive improves PDMS by 7.5, validating the effectiveness of our training paradigm. It also surpasses the prior vision-only state-of-the-art, PARA-Drive (Weng et al., 2024), by 6.8 PDMS.

**Bench2Drive Closed-loop Performance.** Tab. 2 reports closed-loop and multi-ability results on the CARLA Bench2Drive leaderboard. ReCogDrive achieves the highest scenario success rate of 45.45% and the top Driving Score of 71.36, surpassing prior end-to-end baselines. It also excels in safety-critical skills such as emergency braking 69.09% and traffic sign compliance 71.34, while maintaining strong efficiency and a competitive multi-ability mean of 42.03. These results highlight the effectiveness and reliability of our framework in complex urban driving.

**Ablation study on ReCogDrive.** Tab. 3 presents an ablation study on the proposed components of ReCogDrive. When training InternVL3 solely on NAVSIM trajectory data, the model achieves a PDMS of 83.3. Adapting the VLM to driving scenarios with our large-scale driving QA data increases PDMS by 1.7. Introducing the diffusion planner for continuous trajectory prediction further raises PDMS by 2.4. Finally, DiffGRPO achieves a PDMS of 90.8 with a 4.3 improvement, demonstrating the effectiveness of our RL scheme in producing safer driving behavior.

**Comparison of Trajectory Generation Methods.** We compare trajectory generation methods in terms of inference speed, PDM Score, and error rate as shown in Tab. 4. The error rate quantifies the

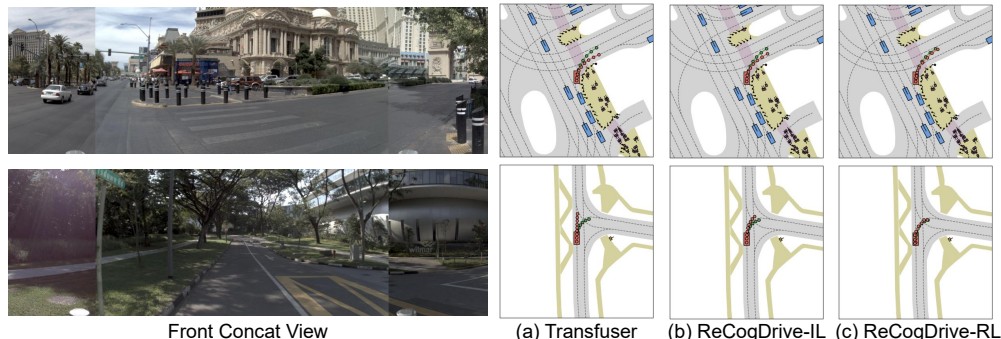

Front Concat View        (a) Transfuser    (b) ReCogDrive-IL   (c) ReCogDrive-RL

Figure 6: **Comparisons on the Navtest benchmark.**

occurrence of malformed outputs that lack a parsable trajectory, which constitutes a low-probability failure mode in direct trajectory regression using VLMs. All inference times in Tab. 4 are measured as end-to-end wall-clock latency for processing a single sample.

Generating trajectories as plain text is slow due to the autoregressive process and may suffer from format errors that compromise safety. In contrast, MLP, Query-based Decoder (Chitta et al., 2022), and diffusion planner are substantially faster, with diffusion-based approaches further boosting PDMS. Building on this, we integrated lightweight architectural refinements including SwiGLU FFN, RMS Norm, RoPE, and QK Norm. Our final model ReCogDrive achieves a $3.5\times$ speedup and a 2.4 PDMS gain over the text baseline, validating the effectiveness of our design.

**Evaluation on Drive VQA Benchmarks.** To provide a comprehensive assessment of our model's language-related capabilities in driving scenarios, we evaluate its performance on the DriveLM and DriveBench benchmarks. The results, presented in Tab. 5, demonstrate the effectiveness of our approach. Our model, ReCogDrive, achieves state-of-the-art performance, surpassing the powerful closed-source GPT-4o model on most key metrics, including the overall DriveBench average score. Furthermore, ReCogDrive significantly outperforms several specialized open-source driving models, validating its strong visual question answering and reasoning abilities in this domain.

**Comparison of RL Algorithms for Diffusion Planning.** To validate DiffGRPO, we compare it with DPPO (Ren et al., 2024) and REINFORCE (Williams, 1992) using the same diffusion planner and PDMS reward, with results in Tab. 6. Since DiffGRPO samples a group of candidate trajectories under the same scene condition, it naturally matches autonomous driving where a single scene often admits multiple safe ground-truth trajectories, and the group-relative advantage encourages safer and more stable rollouts. As reflected by the training curves, DiffGRPO achieves the best PDMS while converging faster and more stably than DPPO and REINFORCE.

**Qualitative Results.** In Fig. 6, we compare ReCogDrive (IL and RL) with Transfuser (Chitta et al., 2022), where RL yields safer and more reliable trajectories in challenging turning scenarios. More visualizations are in the supplementary material.

# 6 CONCLUSION

In this work, we proposed ReCogDrive, a novel reinforced cognitive framework for end-to-end autonomous driving that integrates a Vision-Language Model with a diffusion planner. We first introduced a scalable hierarchical data pipeline that mimics human driving cognition to construct a foundation model with strong driving priors. Building on this foundation, we designed a cognition-guided diffusion planner that injects VLM-derived cognitive tokens into the denoising process, enabling the generation of continuous, stable, and human-like trajectories. To further refine planning behavior, we introduced DiffGRPO, a reinforcement learning scheme that explicitly optimizes safety and comfort in closed-loop driving. Extensive experiments on multiple benchmarks demonstrate that ReCogDrive achieves state-of-the-art performance, validating the effectiveness of our approach. We hope our work can advance the development of autonomous driving VLAs.

## ACKNOWLEDGMENTS AND DISCLOSURE OF FUNDING

This work was partially supported by the National Natural Science Foundation of China under Grant U25B2067.

## ETHICS STATEMENT

We acknowledge and adhere to the ICLR Code of Ethics throughout this work. Our research does not involve human subjects or sensitive personal data, and all experiments were conducted in compliance with ethical research practices.

## REPRODUCIBILITY STATEMENT

We have made extensive efforts to ensure reproducibility. In the supplementary materials, we provide details of our training datasets, including composition and preprocessing steps in Appx. D. We further include hyperparameters and training details in Appx. E. In addition, we provide our model code as part of the supplementary materials to facilitate reproduction of our results.

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

## A    APPENDIX

We organize the supplementary material as follows. First, in Appx. B, we address potential questions that may arise from reading the main text. We then report ReCogDrive's performance on the NAVSIM and DriveBench benchmarks, along with more detailed ablation studies in Appx. C. In Appx. D, we provide details of the training data collection and processing pipeline. Appx. E describes ReCogDrive's training and inference implementation, including all key hyperparameters, and also provides details of the evaluation metrics used on NAVSIM and Bench2Drive. Appx. F and Appx. H discuss the limitations of the method and the broader impacts. We also provide our LLM Usage Statement in Appx. G. Finally, Appx. I presents additional qualitative results, including extensive visualizations on NAVSIM and Bench2Drive, comparisons between IL and RL trajectories, driving dialogues, and analyses of failure cases.

## B    QUESTIONS

**Q1.** *What is the technical novelty in this paper?*

The key novelty of this work is a reinforced cognitive framework for autonomous driving that integrates a VLM with a diffusion-based planner. Unlike prior driving VLA approaches (e.g., EMMA (Hwang et al., 2024), OmniDrive (Wang et al., 2024b)) that generate trajectories purely in the text space, our framework injects VLM-derived driving priors into a diffusion planner, avoiding slow inference, infeasible actions, and trajectory errors. Within this unified framework, we further contribute three innovations: (i) a scalable hierarchical data pipeline that mimics the sequential cognitive process of human drivers through generation, refinement, and quality control, enabling flexible adaptation of VLMs to diverse driving scenarios; (ii) a cognition-guided diffusion planner that leverages VLM-derived priors to efficiently generate continuous and stable trajectories, surpassing plain-text and Transfuser-like (Chitta et al., 2022) decoders; and (iii) Diffusion Group Relative Policy Optimization (DiffGRPO), a reinforcement learning scheme tailored for diffusion-based planners in autonomous driving VLAs, enabling simulation-driven policy optimization for safer and more comfortable driving.

**Q2.** *Does the VLM merely act as a feature extractor?*

ReCogDrive consists of a Vision-Language Model and a diffusion planner. In our framework, the VLM can produce high-level instructions or chain-of-thought reasoning to guide planning. However, as shown in Tab. 11, reinforcement learning alone already achieves state-of-the-art performance on the NAVSIM benchmark. Incorporating CoT-based guidance provides no gain, mainly due to the limited diversity of NAVSIM scenarios. Moreover, Bench2Drive already supplies explicit navigation instructions. For efficiency, we therefore keep VLM-based textual guidance optional.

**Q3.** *Why was the model trained on datasets such as CODA-LM and OmniDrive, yet not evaluated on them?*

Our model is trained on a mixture of 12 open-source driving datasets, which enables handling multiple scenes and tasks. We observed that most existing benchmarks, including CODA-LM and OmniDrive, employ traditional metrics such as BLEU and ROUGE. These metrics evaluate performance by calculating the textual similarity between a model's response and the ground-truth answer. Such metrics primarily assess dataset-specific fitting rather than general reasoning ability. Therefore, to better evaluate the model's true Visual Question Answering (VQA) capabilities, we utilize the GPT-Score from the DriveLM and DriveBench benchmarks, as presented in Tab. 5.

**Q4.** *Does DriveVQA accurately reflect the capabilities of a VLM for autonomous driving?*

DriveBench has shown that many VLMs' decision accuracy does not degrade with visual quality, indicating a reliance on priors rather than genuine visual understanding. Therefore, we primarily use the Planning task to evaluate a model's practical capabilities in dynamic driving scenarios. However, for completeness and to facilitate comparison with existing work, we also provide the VQA results on the DriveLM and DriveBench benchmarks.

**Q5.** *Does ReCogDrive rely on high-quality QA annotations to generalize to other domains?*

ReCogDrive achieves state-of-the-art results by leveraging high-quality driving QA datasets. However, a key contribution of our work is the scalable hierarchical data generation pipeline. Built on advanced VLMs deployed with Sglang, this pipeline can automatically generate large-scale QA annotations, followed by normalization, scoring, and filtering to ensure quality. This design ensures efficiency and adaptability, enabling transfer to new domains without costly manual annotation.

**Q6.** *Why does applying the formula in Eq. 14 to the averaged sub-metrics in Tab. 1 not yield the reported PDM Score?*

As documented in a NAVSIM issue [1], the PDMS score is calculated per scene using the formula and is then averaged across all scenes. Applying the formula to sub-metrics that have already been averaged is a different computation and will thus lead to a different final value.

**Q7.** *Why does the paper adopt such a low discount factor (0.6) for a long-horizon driving task?*

In Eq. 12, the discount factor is applied across denoising steps rather than over the temporal horizon of the driving task. In diffusion-based planning, early denoising steps amplify high-variance noise and often destabilize learning. A smaller discount factor places greater weight on later, more reliable denoising steps, which directly improves stability and feasibility of generated trajectories. This design choice is thus motivated by the dynamics of the denoising process rather than by a short-horizon assumption of the driving task itself.

**Q8.** *Why does the paper employ a BC loss instead of the standard KL-divergence loss?*

We employ a BC-style objective as a stable surrogate for KL regularization. Direct KL constraints often lead to unstable optimization in diffusion-based RL, since variance amplification across denoising steps can cause noisy and unreliable gradients. In contrast, the BC formulation provides smoother updates and better training stability, which proved crucial for policy learning effectiveness.

**Q9.** *Does the diffusion planner only provide performance gains?*

As shown in Tab. 4, beyond the $+2.4$ PDMS improvement over plain-text VLM outputs, the diffusion planner also achieves $3.5\times$ faster inference by requiring only a forward pass rather than auto-regressive generation. More importantly, it eliminates the $0.01\%$ template mismatch errors observed in plain-text decoding, an unacceptable rate in safety-critical driving, making it both more accurate and more reliable for practical autonomous driving.

## C  MORE RESULTS

Table 7: **Performance comparison on Navtest Benchmark with extended metrics.**

| Method | NC↑ | DAC↑ | EP↑ | TTC↑ | C↑ | TL↑ | DDC↑ | LK↑ | EC↑ | EPDMS↑ |
|---|---|---|---|---|---|---|---|---|---|---|
| Transfuser (Chitta et al., 2022) | 97.7 | 92.8 | 79.2 | 92.8 | **100** | 99.9 | 98.3 | 67.6 | 95.3 | 77.8 |
| VADv2 (Chen et al., 2024b) | 97.3 | 91.7 | 77.6 | 92.7 | **100** | 99.9 | 98.2 | 66.0 | 97.4 | 76.6 |
| Hydra-MDP (Li et al., 2024) | 97.5 | 96.3 | 80.1 | 93.0 | **100** | 99.9 | 98.3 | 65.5 | 97.4 | 79.8 |
| Hydra-MDP++ (Li et al., 2024) | 97.9 | **96.5** | 79.2 | 93.4 | **100** | **100** | 98.9 | 67.2 | **97.7** | 80.6 |
| ARTEMIS (Feng et al., 2025) | 98.3 | 95.1 | 81.5 | 97.4 | **100** | 99.8 | 98.6 | **96.5** | 98.3 | 83.1 |
| **ReCogDrive** | **98.3** | 95.9 | **87.7** | **97.7** | 97.9 | 97.4 | **99.3** | 86.0 | 81.9 | **84.2** |

**Experiments on NAVSIM with extended metrics.** Hydra MDP++ (Li et al., 2024) introduces additional evaluation metrics: Traffic Light Compliance (TL), Lane Keeping Ability (LK), Driving Direction Compliance (DDC) and Extended Comfort (EC) to more comprehensively assess driving performance. We evaluate ReCogDrive on NAVSIM using these extended metrics as well. Tab. 7

---

[1]https://github.com/autonomousvision/navsim/issues/116

Table 8: **Evaluation of ReCogDrive-VLM on DriveBench.** We report ReCogDrive-VLM's performance across four key driving dimensions: perception, prediction, planning, and behavior. Results are shown under three input settings: Clean (unaltered images), Corr. (images with averaged corruptions), and T.O. (text-only inputs).

| Method | Size | Type | Perception | | | Prediction | | | Planning | | | Behavior | | |
|---|---|---|---|---|---|---|---|---|---|---|---|---|---|---|
| | | | Clean | Corr. | T.O. | Clean | Corr. | T.O. | Clean | Corr. | T.O. | Clean | Corr. | T.O. |
| Human | - | - | 47.67 | 38.32 | - | - | - | - | - | - | - | 69.51 | 54.09 | - |
| GPT-4o (Hurst et al., 2024) | - | Commercial | 35.37 | 35.25 | 36.48 | 51.30 | 49.94 | 49.05 | 75.75 | 75.36 | 73.21 | 45.40 | 44.33 | 50.03 |
| LLaVA-1.5 (Liu et al., 2023b) | 7B | Open | 23.22 | 22.95 | 22.31 | 22.02 | 17.54 | 14.64 | 29.15 | 31.51 | 32.45 | 13.60 | 13.62 | 14.91 |
| LLaVA-1.5 (Liu et al., 2023b) | 13B | Open | 23.35 | 23.37 | 22.37 | 36.98 | 37.78 | 23.98 | 34.26 | 34.99 | 38.85 | 32.99 | 32.43 | 32.79 |
| LLaVA-NeXT (Liu et al., 2024a) | 7B | Open | 24.15 | 19.62 | 13.86 | 35.07 | 35.89 | 28.36 | 45.27 | 44.36 | 27.58 | 48.16 | 39.44 | 11.92 |
| InternVL2 (Chen et al., 2024d) | 8B | Open | 32.36 | 32.68 | 33.60 | 45.52 | 37.93 | 48.89 | 53.27 | 55.25 | 34.56 | 54.58 | 40.78 | 20.14 |
| Phi-3 (Abdin et al., 2024) | 4.2B | Open | 22.88 | 23.93 | 28.26 | 40.11 | 37.27 | 22.61 | 60.03 | 61.31 | 46.88 | 45.20 | 44.57 | 28.22 |
| Phi-3.5 (Abdin et al., 2024) | 4.2B | Open | 27.52 | 27.51 | 28.26 | 45.13 | 38.21 | 4.92 | 31.91 | 28.36 | 46.30 | 37.89 | 49.13 | 39.16 |
| Oryx (Liu et al., 2024c) | 7B | Open | 17.02 | 15.97 | 18.47 | 48.13 | 46.63 | 12.77 | 53.57 | 55.76 | 48.26 | 33.92 | 33.81 | 23.94 |
| Qwen2-VL (Wang et al., 2024a) | 7B | Open | 28.99 | 27.85 | 35.16 | 37.89 | 39.55 | 37.77 | 57.04 | 54.78 | 41.66 | 49.07 | 47.68 | 54.48 |
| Qwen2-VL (Wang et al., 2024a) | 72B | Open | 30.13 | 26.92 | 17.70 | 49.35 | 43.49 | 5.57 | 61.30 | 63.07 | 53.35 | 51.26 | 49.78 | 39.46 |
| DriveLM (Sima et al., 2024) | 7B | Specialist | 16.85 | 16.00 | 8.75 | 44.33 | 39.71 | 4.70 | 68.71 | 67.60 | 65.24 | 42.78 | 40.37 | 27.83 |
| Dolphins (Ma et al., 2024) | 7B | Specialist | 9.59 | 10.84 | 11.01 | 32.66 | 29.88 | 39.98 | 52.91 | 53.77 | 60.98 | 8.81 | 8.25 | 11.92 |
| ReCogDrive-VLM | 8B | Specialist | 64.95 | 63.35 | 71.44 | 49.34 | 48.92 | 53.49 | 70.20 | 72.5 | 71.76 | 42.36 | 42.76 | 44.00 |

Table 9: Impact of Different BC Loss Weights $\lambda$. Table 10: Impact of Different Min Samplings $\sigma_{\min}^{\exp}$.

| BC Wt. | NC | DAC | TTC | Conf. | EP | PDMS |
|---|---|---|---|---|---|---|
| 0.001 | 97.9 | 97.3 | 94.9 | 100 | 87.3 | 90.8 |
| 0.01 | 97.9 | 97.5 | 94.9 | 100 | 86.0 | 90.3 |
| 0.1 | 97.6 | 97.2 | 93.9 | 100 | 85.0 | 89.3 |

| Min Samp. | NC | DAC | TTC | Conf. | EP | PDMS |
|---|---|---|---|---|---|---|
| 0.01 | 98.2 | 97.8 | 95.6 | 100 | 82.8 | 89.5 |
| 0.02 | 97.9 | 97.3 | 94.9 | 100 | 87.3 | 90.8 |
| 0.04 | 97.9 | 97.7 | 95.3 | 100 | 85.7 | 90.5 |

Table 11: Comparison between trajectory-only (Only Traj), adding high-level command (w/ High Com.), and chain-of-thought (w/ CoT).

Table 12: Impact of different discount factors $\gamma$. We use discounting to reduce the influence of early-step noise during the diffusion process.

| Mode | NC | DAC | TTC | Conf. | EP | PDMS |
|---|---|---|---|---|---|---|
| Only Traj | 97.9 | 97.3 | 94.9 | 100 | 87.3 | 90.8 |
| w/ High Com. | 97.9 | 97.3 | 94.9 | 100 | 87.2 | 90.8 |
| w/ CoT | 97.8 | 97.2 | 94.8 | 100 | 87.1 | 90.7 |

| Discount | NC | DAC | TTC | Conf. | EP | PDMS |
|---|---|---|---|---|---|---|
| 0.6 | 97.9 | 97.3 | 94.9 | 100 | 87.3 | 90.8 |
| 0.8 | 98.0 | 97.6 | 95.3 | 100 | 85.9 | 90.5 |
| 1.0 | 97.9 | 97.5 | 94.9 | 100 | 86.0 | 90.3 |

compares our approach against existing methods under this metrics. ReCogDrive achieves state of the art scores in Driving Direction Compliance (DDC), No at-fault Collisions (NC), Ego Progress (EP) and Time to Collision (TTC), and delivers a 1.1 improvement in EPDMS over ARTEMIS (Feng et al., 2025), demonstrating the effectiveness of our method.

**Evaluations of VLMs across driving tasks.** Tab. 8 reports results on *DriveBench* across perception, prediction, planning, and behavior. ReCogDrive-VLM achieves the best overall performance and shows strong robustness under both corrupted and text-only settings, outperforming general-purpose VLMs (e.g., Qwen2-VL, InternVL2) and specialist models (e.g., DriveLM, Dolphins). These results demonstrate the strong driving scene understanding capability of our ReCogDrive-VLM.

**Impact of Different Behavior Clone Loss Weights.** Tab. 9 examines the impact of the BC loss weight $\lambda$. As $\lambda$ decreases, the policy learns more aggressively, and at $\lambda = 0.001$ ego progress increases to 87.3 with slight declines in NC, DAC, and TTC, yielding the highest PDMS.

**Impact of Different Min Sampling Values.** Following (Ren et al., 2024; Nichol & Dhariwal, 2021), we apply a cosine schedule to the diffusion noise variances $\sigma_k$ and clip them to a minimum value $\sigma_{\min}^{\exp}$. Clipping $\sigma_k$ to a nonzero minimum encourages the diffusion policy to sample more diverse trajectories, yet overly large $\sigma_{\min}^{\exp}$ can destabilize training. Setting $\sigma_{\min}^{\exp} = 0.02$ achieves a PDMS of 90.8, as shown in Tab. 10.

**Effect of VLM guidance modes.** As shown in Tab. 11, outputting only trajectory achieves performance comparable to adding high-level command guidance, with almost identical PDMS scores. Interestingly, incorporating chain-of-thought reasoning does not further improve the results; instead, it slightly decreases the PDMS score by 0.1. This suggests that the current NAVSIM benchmark

Table 13: **Planning performance on the nuScenes val set.** We train our model on Bench2Drive, and report both zero-shot transfer results on nuScenes and performance after fine-tuning on nuScenes.

| Method | L2 (m)↓ | | | | Collision (%)↓ | | | |
|---|---|---|---|---|---|---|---|---|
| | 1s | 2s | 3s | Avg. | 1s | 2s | 3s | Avg. |
| UniAD (Hu et al., 2023) | 0.20 | 0.42 | 0.75 | 0.46 | 0.02 | 0.25 | 0.84 | 0.37 |
| VLP (Pan et al., 2024) | 0.30 | 0.53 | 0.84 | 0.55 | 0.01 | 0.07 | 0.38 | 0.15 |
| VAD (Jiang et al., 2023) | 0.17 | 0.34 | 0.60 | 0.37 | 0.07 | 0.10 | 0.24 | 0.14 |
| DriveVLM (Tian et al., 2024) | 0.18 | 0.34 | 0.68 | 0.40 | 0.10 | 0.22 | 0.45 | 0.27 |
| DiMA-Dual+ (Hegde et al., 2025) | 0.14 | 0.27 | 0.46 | 0.29 | 0.05 | 0.07 | 0.15 | 0.09 |
| AutoVLA (Zhou et al., 2025b) | 0.22 | 0.39 | 0.61 | 0.41 | 0.10 | 0.17 | 0.28 | 0.18 |
| ReCogDrive (zero-shot) | 1.34 | 1.91 | 2.69 | 1.98 | 0.07 | 0.57 | 0.82 | 0.82 |
| ReCogDrive (fine-tune) | 0.17 | 0.34 | 0.63 | 0.38 | 0.07 | 0.09 | 0.16 | 0.10 |

Table 14: Sensitivity of PDMS to the EP weight $w_{\text{EP}}$ (fixed $w_{\text{TTC}} = 5.0$, $w_{\text{C}} = 2.0$).

| $w_{\text{EP}}$ | NC | DAC | TTC | Conf. | EP | PDMS |
|---|---|---|---|---|---|---|
| 5.0 | 98.6 | 97.8 | 96.2 | 100 | 84.0 | 90.3 |
| 10.0 | 97.9 | 97.3 | 94.9 | 100 | 87.3 | 90.8 |
| 20.0 | 97.8 | 97.3 | 94.2 | 100 | 87.3 | 90.5 |
| 30.0 | 97.6 | 97.2 | 93.1 | 99.6 | 87.6 | 90.0 |

Table 15: Sensitivity of PDMS to the TTC weight $w_{\text{TTC}}$ (fixed $w_{\text{EP}} = 10.0$, $w_{\text{C}} = 2.0$).

| $w_{\text{TTC}}$ | NC | DAC | TTC | Conf. | EP | PDMS |
|---|---|---|---|---|---|---|
| 5.0 | 97.9 | 97.3 | 94.9 | 100 | 87.3 | 90.8 |
| 10.0 | 98.5 | 97.7 | 95.9 | 100 | 85.1 | 90.5 |
| 20.0 | 98.6 | 97.7 | 96.2 | 100 | 84.5 | 90.5 |
| 30.0 | 98.5 | 97.8 | 96.1 | 100 | 84.5 | 90.4 |

lacks sufficient diversity to demonstrate the benefits of high-level command and CoT reasoning, so we keep these guidance optional.

**Impact of Different Discount Factors.** Tab. 12 shows the impact of the discount factor $\gamma$ on RL fine-tuning. When $\gamma = 1.0$, all timesteps, including the very noisy early steps, contribute equally to the policy gradient, which amplifies high variability noise and destabilizes learning. In contrast, setting $\gamma = 0.6$ focuses the update on later, more reliable steps and achieves the best PDMS of 90.8.

**Open-loop evaluation on nuScenes.** We report planning results on the nuScenes val set in Tab. 13. Trained on Bench2Drive, our method achieves strong zero-shot performance on nuScenes, and fine-tuning further yields excellent results. In particular, fine tuning substantially lowers collision rates, outperforming several prior planners, while maintaining competitive L2 displacement errors across all horizons.

**PDM Score Coefficient Sensitivity Analysis.** Our PDMS reward is constructed from five components: EP, TTC, Comfort (C), NC, and DAC. Concretely, the PDMS is computed as:

$$\text{PDMS} = \text{NC} \times \text{DAC} \times \left( \frac{w_{\text{EP}} \cdot \text{EP} + w_{\text{TTC}} \cdot \text{TTC} + w_{\text{C}} \cdot \text{C}}{w_{\text{EP}} + w_{\text{TTC}} + w_{\text{C}}} \right). \tag{13}$$

Tabs. 14 and 15 present the sensitivity of PDMS to the coefficients $w_{\text{EP}}$ and $w_{\text{TTC}}$ respectively. Increasing the EP weight $w_{\text{EP}}$ makes the policy more aggressive, meaning that EP rises while NC, DAC, and TTC decrease slightly, and the overall PDMS correspondingly decreases when is set too large. In contrast, increasing the TTC weight $w_{\text{TTC}}$ results in safer time-to-collision behavior with higher TTC but reduced EP, which leads to diminishing PDMS improvements as $w_{\text{TTC}}$ increases. Based on these results, we choose the configuration $w_{\text{EP}} = 10.0$, $w_{\text{TTC}} = 5.0$, and $w_{\text{C}} = 2.0$ as our PDMS reward setting, since it provides the best trade-off between progress and safety.

## D  TRAINING DATASETS CONSTRUCTION

### D.1  DATA COLLECTION

We collect 12 open-source driving QA datasets, including Talk2Car (Deruyttere et al., 2019), SUTD (Xu et al., 2021), NuScenes-QA (Qian et al., 2024), OmniDrive (Wang et al., 2024b), and others, yielding over 3.1 million question-answer pairs that cover perception, prediction, and planning tasks across diverse real-world scenarios.

**Talk2car** (Deruyttere et al., 2019) is built on the nuScenes (Caesar et al., 2020) dataset and contains 850 videos from the nuScenes (Caesar et al., 2020) training set. This dataset has 11,959 natural language commands.

**SUTD** (Xu et al., 2021) contains 10,080 in-the-wild videos and annotated 62,535 QA pairs. These videos are obtained through a combination of online collection and offline shooting, covering various weather conditions, times, and road conditions.

**DRAMA** (Malla et al., 2023) is a dataset collected to investigate risk location and natural language description in driving scenarios. It contains 17,785 interactive driving scenarios.

**NuScenes-QA** (Qian et al., 2024) encompasses 34,149 complex autonomous driving scenes and 459,941 question-answer pairs, with various types of questions. It aims to evaluate a model's ability to understand and reason about complex visual data in multi-modal, multi-frame, and outdoor scenarios.

**DriveGPT4** (Xu et al., 2024) is built based on the BDD-X (Kim et al., 2018) dataset, which contains about 20,000 samples. By dividing them into 16,803 training segments and 2,123 testing segments, question-answer pairs are generated using the control signal data and text annotations.

**LingoQA** (Marcu et al., 2024) contains 28K unique short video scenarios and 419K annotations. The dataset covers various questions related to driving scenarios, including aspects such as behaviors, scenery, and perception.

**DriveLM** (Sima et al., 2024) consists of a training set of 4,072 frames and a validation set of 799 frames, with an average of 91.4 QA pairs per frame.

**MAPLM** (Cao et al., 2024) contains point-cloud BEV projections and surround view images of various traffic scenes, such as highways and urban roads, and is equipped with element-level, lane-level, and road-level scene descriptions.

**NuInstruct** (Ding et al., 2024) is a dataset constructed based on Nuscenes (Caesar et al., 2020), containing 91K multi-view video instruction-response pairs in 17 subtasks.

**CODA-LM** (Chen et al., 2024a) comprises 9,768 real-world driving scenarios with 41,722 textual annotations for critical road entities and 21,537 annotations for road corner cases.

**OminiDrive** (Wang et al., 2024b) covers 3D perception, reasoning, and planning tasks, including offline and online question-response tasks.

**Senna** (Jiang et al., 2024a) design a series of planning-oriented QAs including scene description, traffic signal detection, vulnerable road user identification, motion intention prediction, meta-action planning and planning explanation. Since the senna dataset is not publicly available, we used Qwen2.5-VL-72B (Bai et al., 2025) for question-answer data annotation.

### D.2  SCALABLE HIERARCHICAL DATA PIPELINE

To systematically enhance VLM performance for autonomous driving, we design a scalable hierarchical data pipeline consisting of three stages: *Generation*, *Refinement*, and *Quality Control*, as shown in Fig. 2. This design allows us to construct diverse, semantically consistent, and high-quality annotations at scale.

**Generation.** We first leverage Qwen2.5-VL (Bai et al., 2025) with crafted prompts to generate annotations across the full spectrum of autonomous driving tasks on NAVSIM (Dauner et al., 2025). These tasks span perception (e.g., scene description, key object identification, road marking recognition, traffic light classification, vulnerable road user detection), prediction (e.g., motion prediction), planning (e.g., high-level command prediction and decision explanation), and advanced reasoning (e.g.,

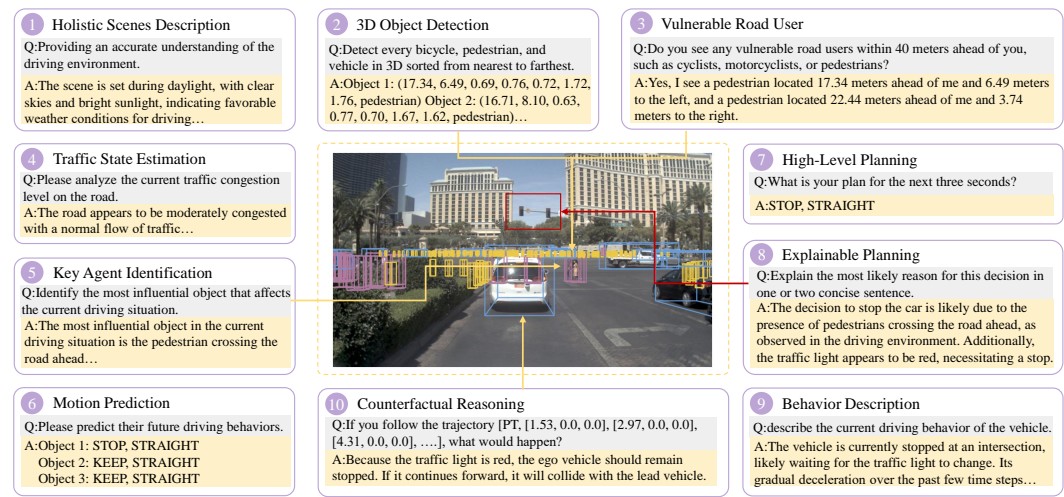

Figure 7: **Example of data generated by our Scalable Hierarchical Data Pipeline.** The pipeline produces structured QA annotations that cover four levels of autonomous-driving capability, from *Foundational Perception* and *Dynamic Understanding* to *Planning and Reasoning* and *Advanced Reasoning*, as illustrated by the multi-task QA set in this scene.

counterfactual scenarios inspired by OmniDrive (Wang et al., 2024b)). To strengthen supervision, we further incorporate NAVSIM's existing sensor and trajectory labels into the generation process. Applying this stage yields a large pool of candidate QA pairs.

**Refinement.** Since open-source datasets employ heterogeneous formats, we unify them through normalization and linguistic enrichment. (i) *Data Normalization:* Different datasets adopt different bounding-box styles. For example, DriveLM (Sima et al., 2024) represents an object as "<c2,CAM_BACK,864.2,468.3>", while NuInstruct (Ding et al., 2024) uses "<car>[c2,584,478,603,516]". We convert all such variants into a standardized tag format: "<car><FRONT_VIEW><box>[$x_1, y_1, x_2, y_2$]</box>" following the InternVL3 (Zhu et al., 2025) pre-training convention. Coordinates $x_1, y_1, x_2, y_2$ are rescaled to the integer range [0,1000] relative to image resolution. (ii) *Data Augmentation:* To enrich linguistic diversity, especially for datasets with limited QA templates (e.g., CODA-LM with only three formats, DRAMA with terse answers), we employ a two-step process. First, an LLM paraphrases question templates to expand structural variation. Second, Qwen2.5-VL is prompted with the original image, question, and answer to generate more detailed, natural responses. This refinement ensures consistency in structure and greater variety in language.

**Quality Control.** Finally, we implement a rigorous filtering step. Qwen2.5-VL assigns each QA pair a quality score based on predefined linguistic and semantic criteria. Any pair scoring below 60 is discarded. After this filtering process, we retain 2.3M high-quality QA pairs. Notably, applying the pipeline to NAVSIM produces 752K well-structured pairs, which we use to fine-tune the VLM, substantially enhancing its planning and reasoning capabilities.

### D.3   DATASET STATISTICS

The dataset consists of 41% perception samples, 11% prediction samples, and 24% planning samples, aimed at improving VLMs' understanding of driving scenarios. We also include 24% visual instruction samples to maintain the model's instruction-following ability. For NAVSIM training, ReCogDrive is trained using a mixture of NAVSIM trajectory data (5%), our newly generated 752K cognition–aligned QA pairs (22.9%), open–source driving QA datasets totaling 2.3M samples (68.1%), and 665K LLaVA instruction data (3.9%). This composition balances driving–specific reasoning with general VLM instruction capability. For Bench2Drive adaptation, the NAVSIM–trained model is further fine–tuned on Bench2Drive execution trajectories (79.2%) together with Bench2Drive driving–QA annotations (20.8%), enabling the model to acquire simulation–specific driving priors required for closed–loop CARLA evaluation.

# E    EXPERIMENT DETAILS

In this section, we first introduce the evaluation metrics of NAVSIM and Bench2Drive, and then detail our model configuration, the hyperparameters used in the three-stage training and evaluation, and our hardware setup.

**Evaluation Metrics.** We evaluate our method on two primary benchmarks. For the planning-oriented NAVSIM benchmark, we use the official Predictive Driver Model Score (PDMS), a closed-loop metric that holistically assesses safety (NC, DAC, TTC), comfort (C), and progress (EP):

$$\text{PDMS} = \text{NC} \times \text{DAC} \times \left( \frac{5 \cdot \text{EP} + 5 \cdot \text{TTC} + 2 \cdot \text{C}}{12} \right). \tag{14}$$

For the CARLA-based Bench2Drive benchmark, which focuses on safety-critical scenarios, we report the Success Rate (SR) and the average Driving Score (DS). The DS combines Route Completion (RC) with multiplicative penalties for infractions (IS):

$$\text{SR} = \frac{N_{\text{success}}}{N_{\text{total}}}; \quad \text{DS} = \frac{1}{N_{\text{total}}} \sum_{i=1}^{N_{\text{total}}} \left( \text{RC}_i \times \prod_j \text{IS}_{i,j} \right). \tag{15}$$

**Model Architecture and Hyperparameter Details.** Our model architecture consists of two main components: Vision-Language Models (VLMs) and a diffusion-based trajectory planner. For the Vision-Language Models, we utilize the pre-trained InternVL3-8B and InternVL3-2B (Zhu et al., 2025) model. The model processes visual inputs by splitting images into patches: on NAVSIM, we use a single front-view image divided into 9 patches, while on Bench2Drive we use multi-view inputs from six cameras, each cropped into 3 patches. The input images are cropped into 448x448 pixel patches. For the diffusion planner, we use the DiT (Peebles & Xie, 2023) architecture, employing DDPM/DDIM (Ho et al., 2020) denoising methods. During training, the denoising process involves 100 steps, while for inference, the diffusion process is reduced to 5 steps. The hidden layer size of the DiT architecture is set to 384, with a head size of 8 and 16 layers.

Table 16: **Hyperparameters for ReCogDrive.**

| Stage | Parameter | Value |
|---|---|---|
| I | Number of epochs | 3 |
| | Batch size | 1024 |
| | Learning rate | $4e^{-5}$ |
| | Weight decay | 0.05 |
| | Warmup ratio | 0.10 |
| | Learning rate schedule | Cosine |
| II | Number of epochs | 200 |
| | Batch size | 512 |
| | Learning rate | $1e^{-4}$ |
| | Learning rate schedule | Cosine |
| | Weight decay | $1 \times 10^{-4}$ |
| III | Number of epochs | 10 |
| | Batch size | 128 |
| | Learning rate | $1e^{-4}$ |
| | Learning rate schedule | Cosine |
| | BC loss weight | $1e^{-2}$ |
| | Denoised clip threshold | $\pm 1.0$ |
| | Noise clip threshold | $\pm 3.0$ |
| | Minimum denoising std | 0.02 |
| | Minimum log-variance std | 0.10 |
| | Discount factor | 0.6 |

**Training Configuration.** In the first stage, we fine-tune the InternVL3 VLM for three epochs with all parameters unfrozen and a batch size of 1,024 on a curated 3M real-world driving QA corpus

that unifies Talk2Car (Deruyttere et al., 2019), SUTD (Xu et al., 2021), DRAMA (Malla et al., 2023), NuScenes-QA (Qian et al., 2024), DriveGPT4 (Xu et al., 2024), LingoQA (Marcu et al., 2024), DriveLM (Sima et al., 2024), MAPLM (Cao et al., 2024), NuInstruct (Ding et al., 2024), CODA-LM (Chen et al., 2024a), OmniDrive (Wang et al., 2024b), Senna (Jiang et al., 2024a), LLaVA Instruction-QA (Liu et al., 2023a) and 752K additional QA pairs generated by our hierarchical data pipeline, providing a strong real-world foundation tailored for both DriveBench and NAVSIM. We use AdamW with a base learning rate of $4e^{-5}$, weight decay of 0.05, and a cosine learning rate schedule with a 10% linear warmup. For CARLA-based simulation, we further adapt the real-world–trained model by fine-tuning it with Bench2Drive trajectory data and Bench2Drive-QA using identical hyperparameters to ensure compatibility with the CARLA environment.

In the second stage, we train the diffusion-based planner with imitation learning on trajectory supervision from both NAVSIM and Bench2Drive for 200 epochs and a batch size of 512, using AdamW with a learning rate that warms up to $1e^{-4}$ over the first 1.5% of steps, then decays cosinely to $1e^{-6}$, and a weight decay of $1e^{-4}$.

In the third stage, we perform DiffGRPO reinforcement learning using the full NAVSIM and Bench2Drive training data for 10 epochs with a batch size of 128, using AdamW and a cosine schedule that decays the learning rate from $1e^{-4}$ to 0. Additionally, we stabilize diffusion training and inference by clipping the reconstructed clean estimate $x_0$ to $\pm1.0$, clipping Gaussian noise samples to $\pm5.0$, enforcing a nonzero floor on the denoising standard deviation $\sigma_t$ of 0.02 for training, clamping $\sigma_t$ to at least 0.10 when computing $\log p(x_{t-1} \mid x_t)$, and applying a discount factor $\gamma = 0.6$ during RL fine-tuning to downweight early, noisy timesteps. During RL fine-tuning, we optimize the planner using the PDMS reward with weighting coefficients $\alpha = 10.0$ (EP), $\beta = 5.0$ (TTC), and $\gamma = 2.0$ (Comfort), which we identified as providing the best trade-off between progress and safety.

**Hardware Configuration.** We implement ReCogDrive on Debian with PyTorch, training across four nodes—each equipped with an Intel® Xeon® Platinum 8457C CPU and eight NVIDIA H20 GPUs (32 GPUs in total). Inference is performed on a single node with 8 GPUs.

## F    LIMITATIONS

Although ReCogDrive achieves state of the art performance on the NAVSIM benchmark, it still faces several limitations. These include the difficulty of processing multiple camera inputs, handling video frame sequences, and relatively high inference latency. Future work may address these issues by designing a 3D vision encoder that aligns with textual features and developing more efficient model architectures. We also plan to deploy and evaluate our model on real vehicles.

## G    LLM USAGE STATEMENT

Large Language Models (LLMs) were employed solely as writing assistants to improve the clarity, grammar, and readability of the manuscript. They were not involved in research ideation, methodology design, data analysis, or result interpretation.

## H    BROADER IMPACTS

Our research promotes the application of Vision-Language Models (VLMs) in the autonomous driving domain. Through simulator-assisted reinforcement learning, our model more effectively mitigates collision risk and generalizes to rare, challenging scenarios. This advancement could lead to safer, more reliable autonomous systems capable of handling real-world driving scenarios.

## I    QUALITATIVE RESULTS

We first present extensive qualitative visualizations of ReCogDrive on NAVSIM and Bench2Drive, highlighting its ability to produce safe and smooth trajectories. Next, we provide trajectory comparisons between ReCogDrive-IL and ReCogDrive-RL to demonstrate the effectiveness of our proposed DiffGRPO. We further showcase dialogue examples generated by ReCogDrive, illustrating its cog-

nitive reasoning capability. Finally, we analyze representative failure cases on NAVSIM to provide insights into current limitations.

## I.1 QUALITATIVE RESULTS ON NAVTEST SPLITS

We present representative cases from the Navtest splits, highlighting ReCogDrive's ability to follow navigation instructions while ensuring safety and smoothness (see Fig. 8 and Fig. 11).

## I.2 QUALITATIVE RESULTS ON BENCH2DRIVE TEST SCENARIOS

We evaluate ReCogDrive on challenging scenarios from Bench2Drive, demonstrating its robustness under diverse traffic conditions and rare corner cases, while consistently maintaining strong performance in closed-loop settings.

## I.3 COMPARISON BEFORE AND AFTER REINFORCEMENT LEARNING

Representative qualitative comparisons are shown in Fig. 12–16, where we contrast Transfuser, imitation learning (IL), and reinforcement learning (RL) variants of ReCogDrive, demonstrating the progressive improvements in decision making, trajectory stability, and safety awareness.

## I.4 FAILURE CASES

We also present representative failure cases of ReCogDrive on the Navsim benchmark, as shown in Fig. 17. These include instances of aggressive driving caused by the model's tendency to prioritize ego progress, leading to assertive maneuvers. In addition, the relatively weak perception capability of current VLMs means that directly feeding multi-view camera inputs sometimes results in suboptimal performance, particularly in turning scenarios. We also observe unsafe following distances due to insufficient modeling of surrounding vehicle behaviors, reflecting the limitations of VLMs in motion prediction. Finally, we identify cases where predicted trajectories are visually close to the ground truth but still scored as failures, suggesting that the evaluation metrics may suffer from false negatives.

## I.5 MODEL–HUMAN DIALOGUE

As shown in Fig. 18–20, ReCogDrive engages in interactive dialogue by providing detailed scene descriptions, recognizing traffic signs, reasoning about driving intent, and generating safe turning trajectories.

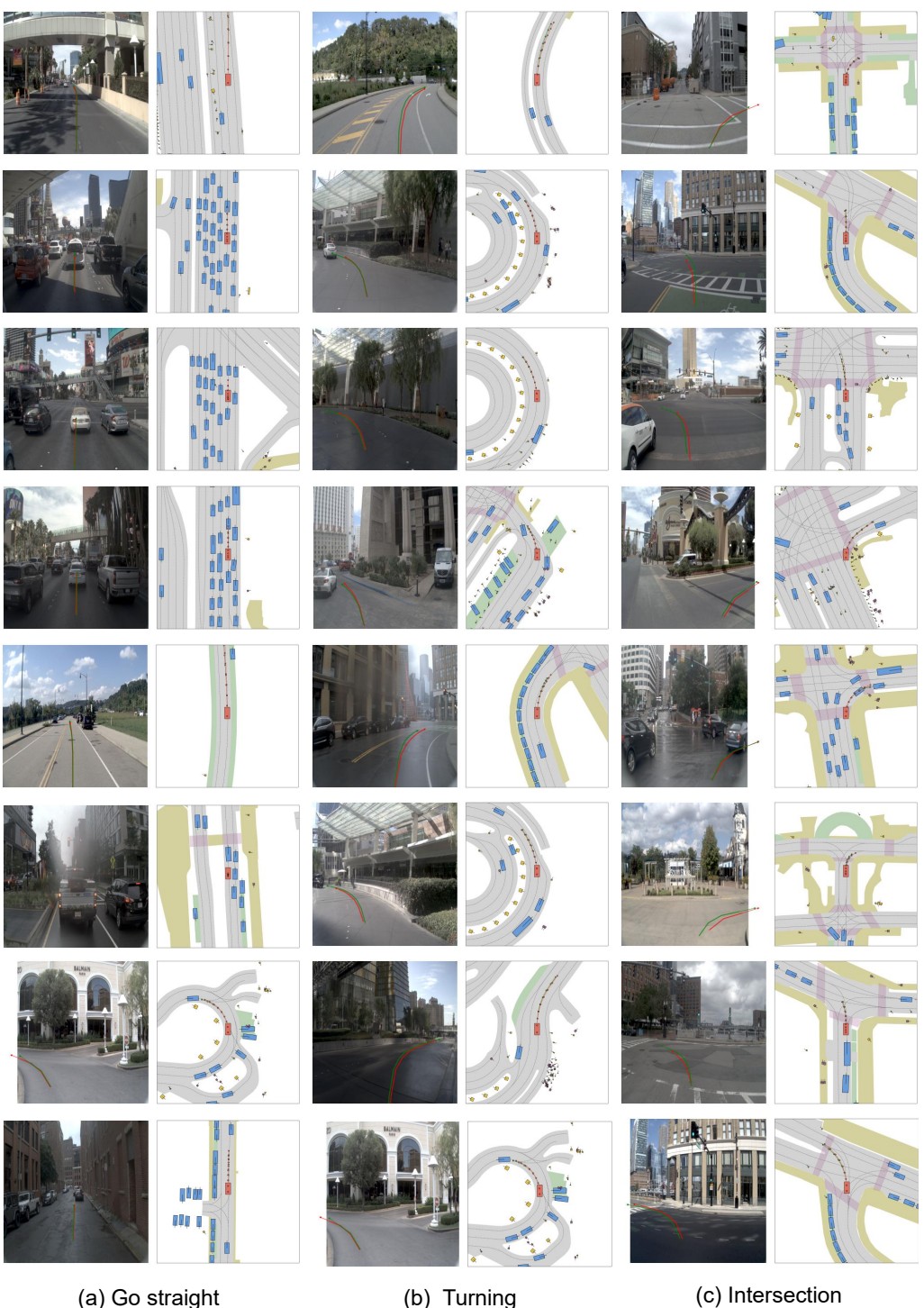

(a) Go straight      (b) Turning      (c) Intersection

Figure 8: Qualitative results on the Navtest benchmark.

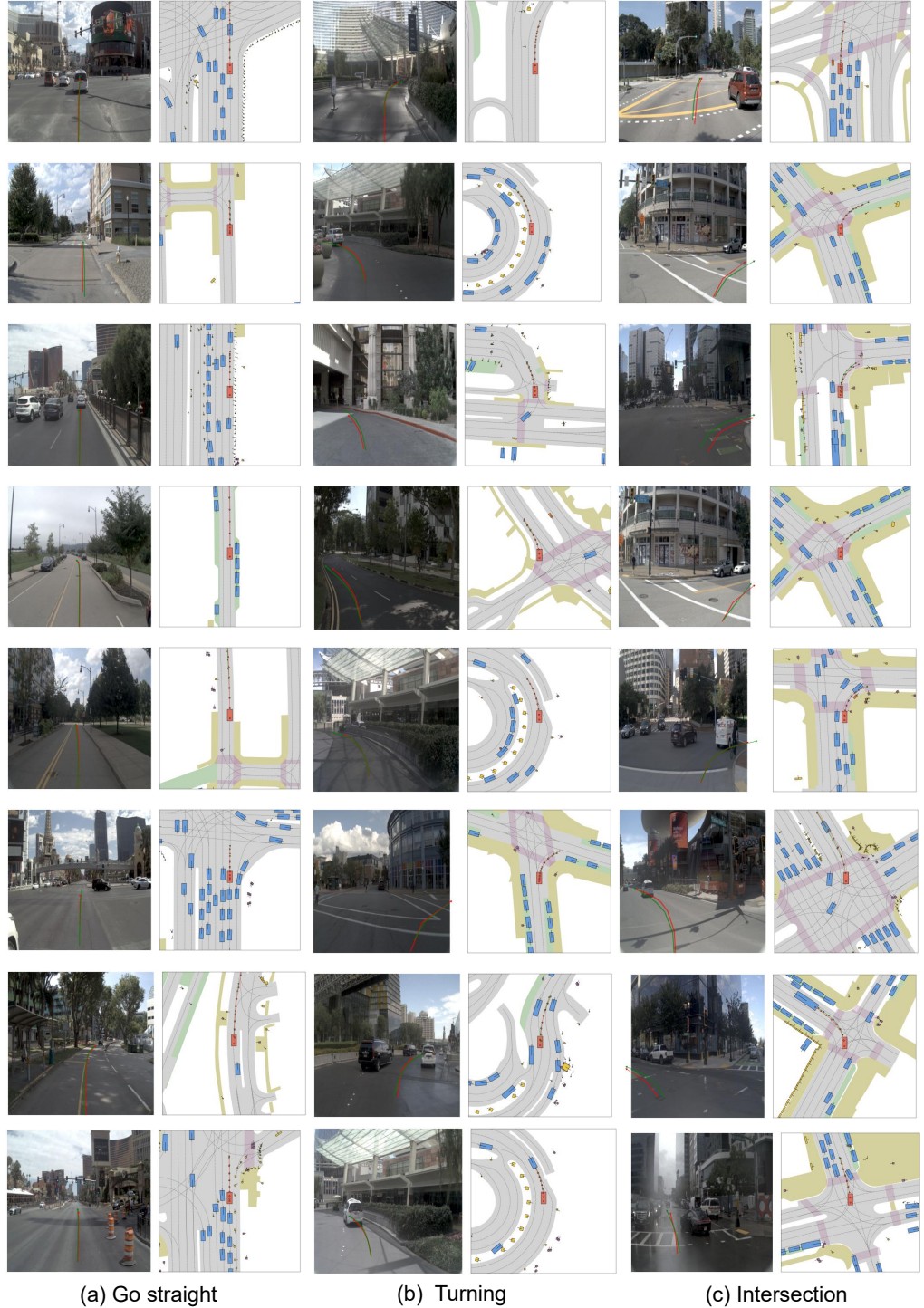

(a) Go straight          (b) Turning          (c) Intersection

Figure 9: Qualitative results on the Navtest benchmark.

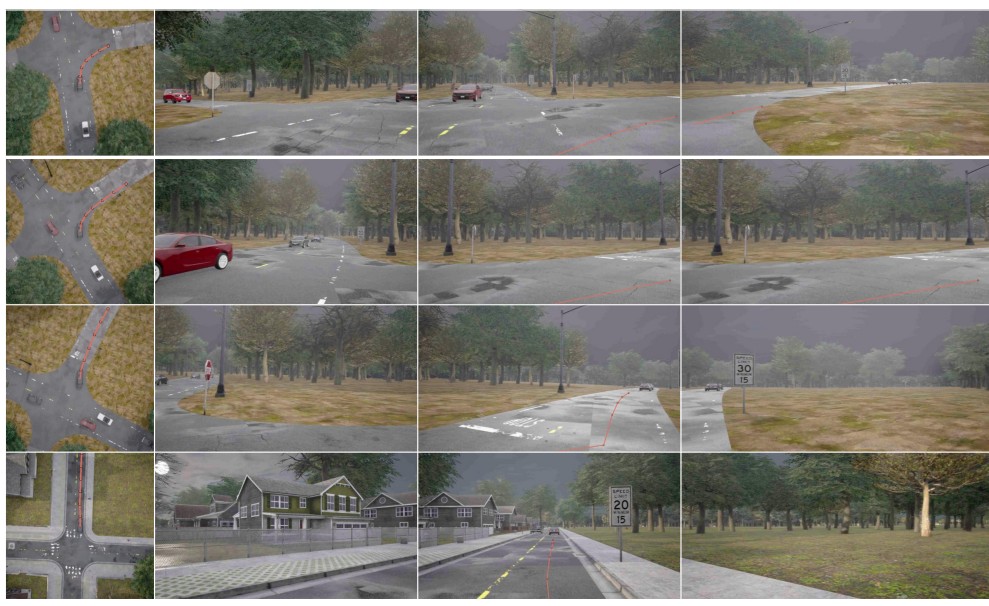

(a) Turning Right

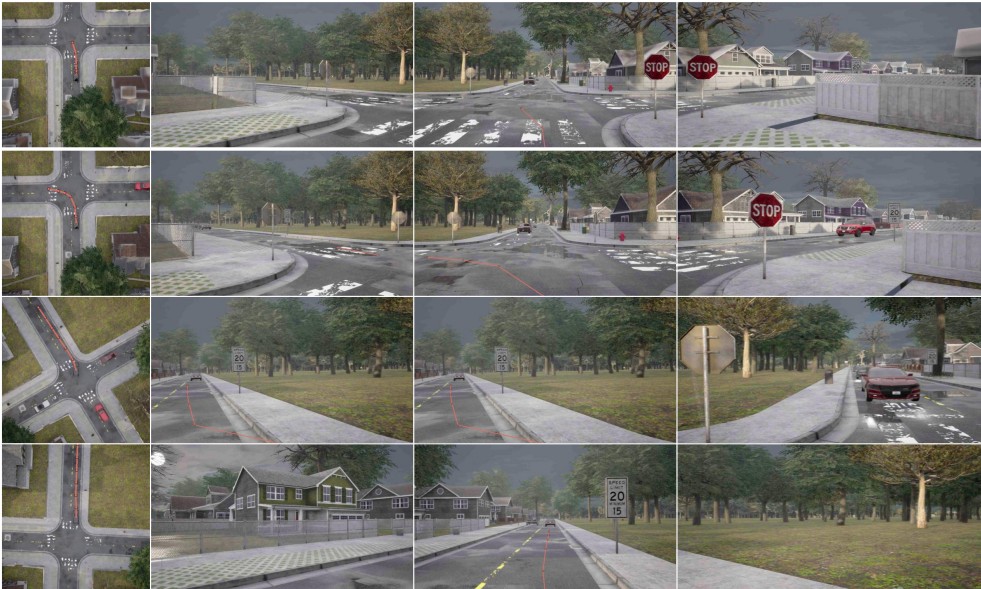

(b) Turning Left

Figure 10: Qualitative results on the Bench2drive benchmark.

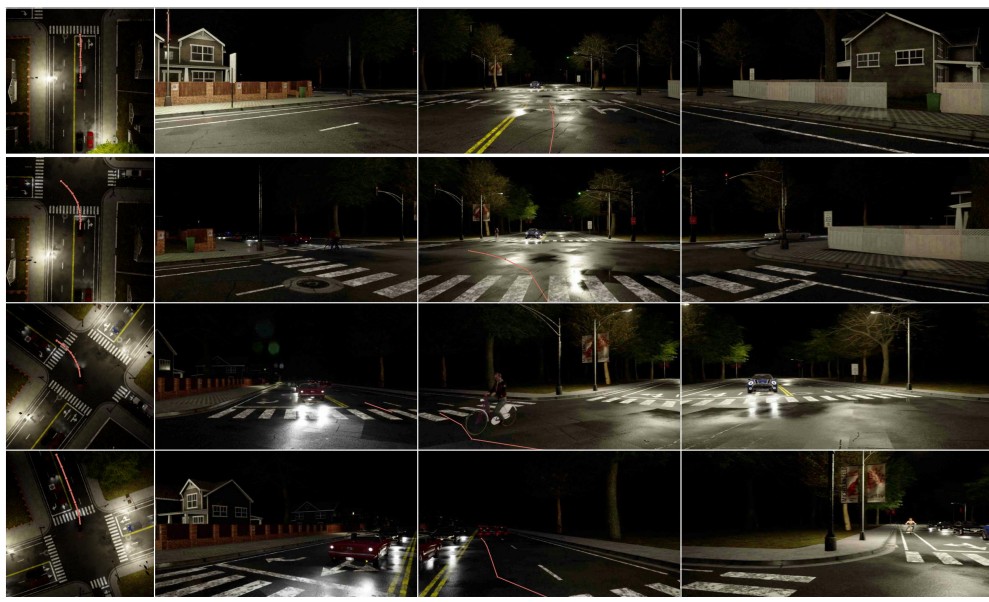

(c) Turning Left At Night

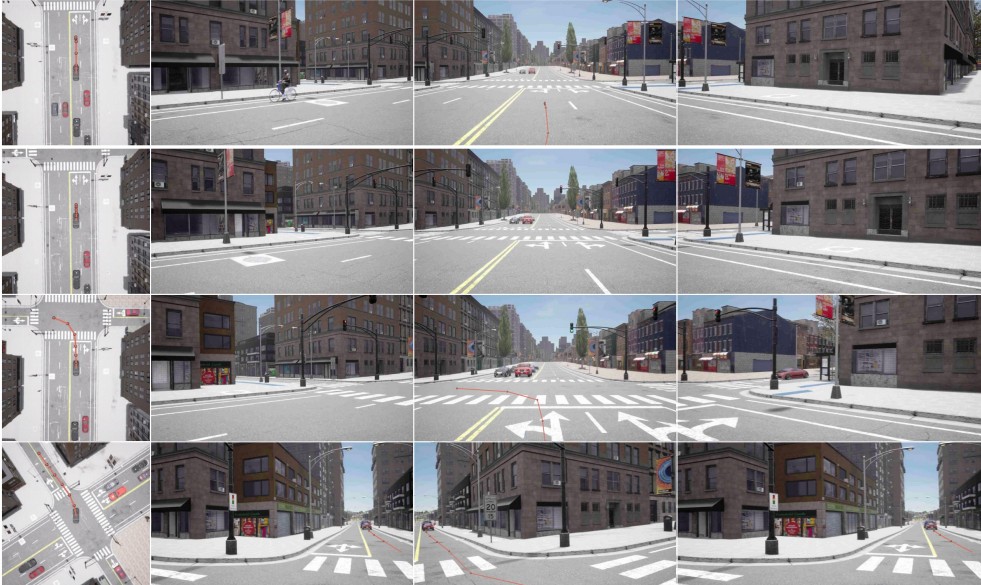

(d) Stop On Red, Proceed On Green.

Figure 11: Qualitative results on the Bench2drive benchmark.

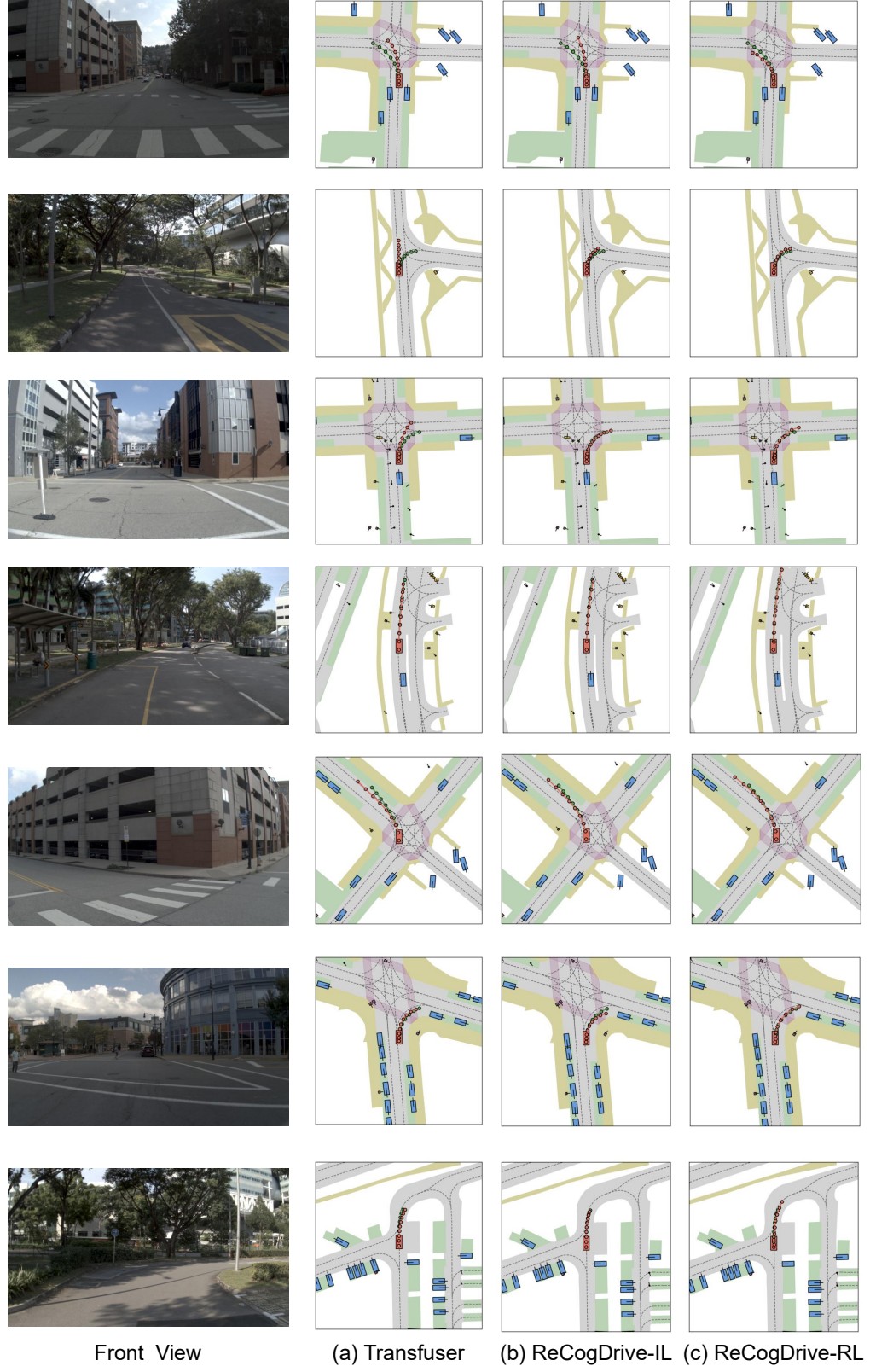

Front View      (a) Transfuser      (b) ReCogDrive-IL    (c) ReCogDrive-RL

Figure 12: Qualitative comparisons on the Navtest benchmark.

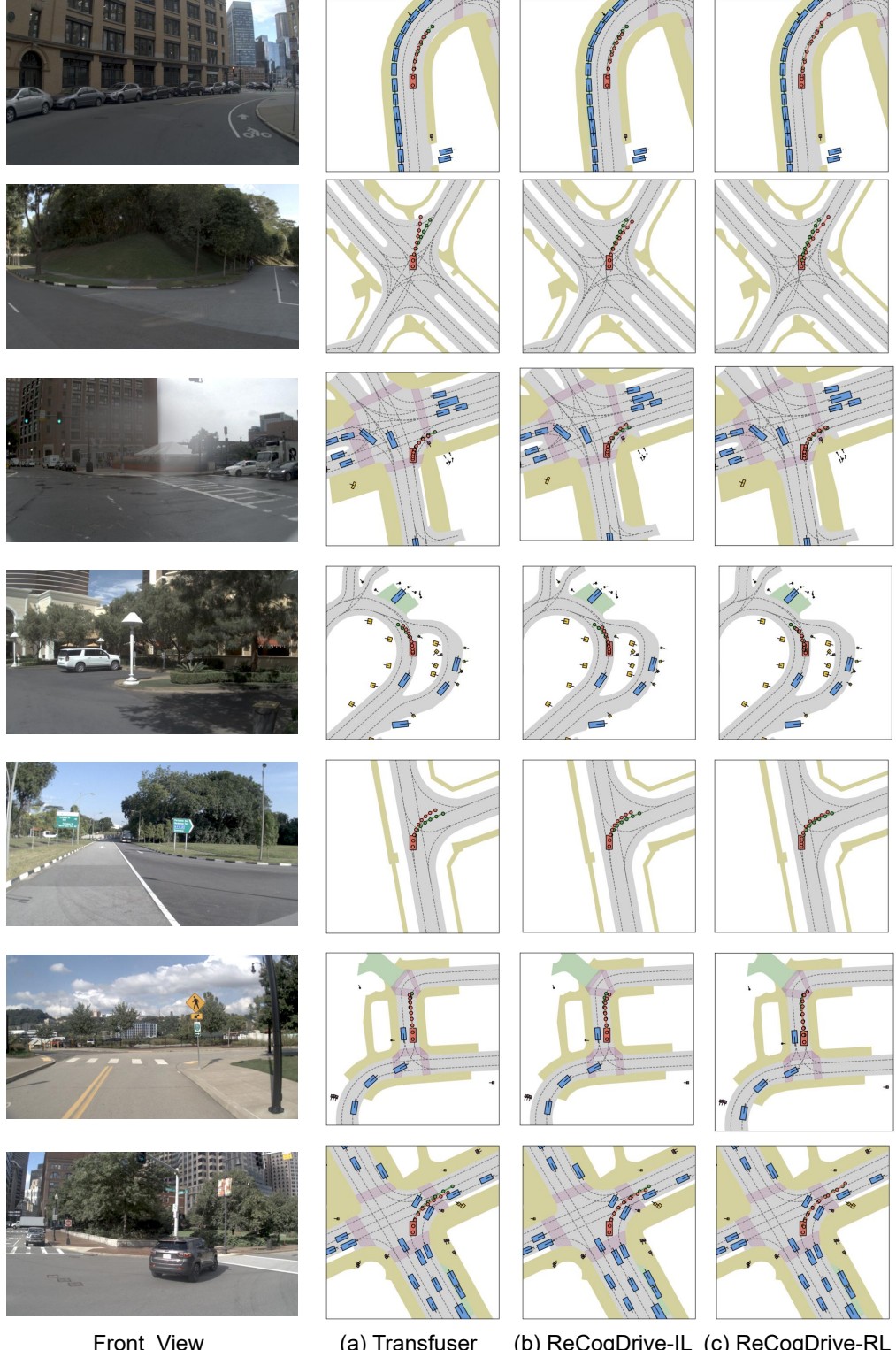

Front View      (a) Transfuser     (b) ReCogDrive-IL    (c) ReCogDrive-RL

Figure 13: Qualitative comparisons on the Navtest benchmark.

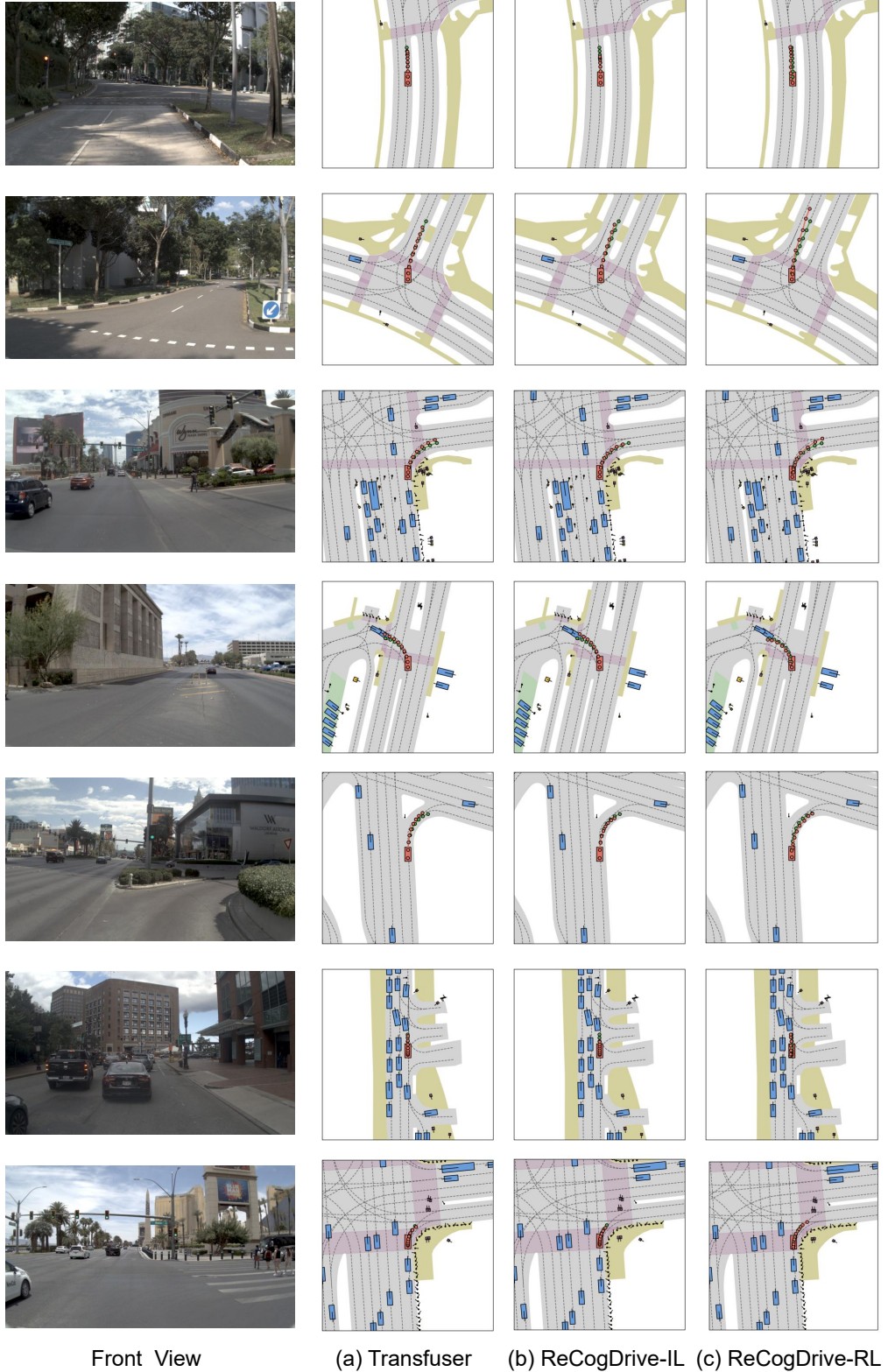

Figure 14: Qualitative comparisons on the Navtest benchmark.

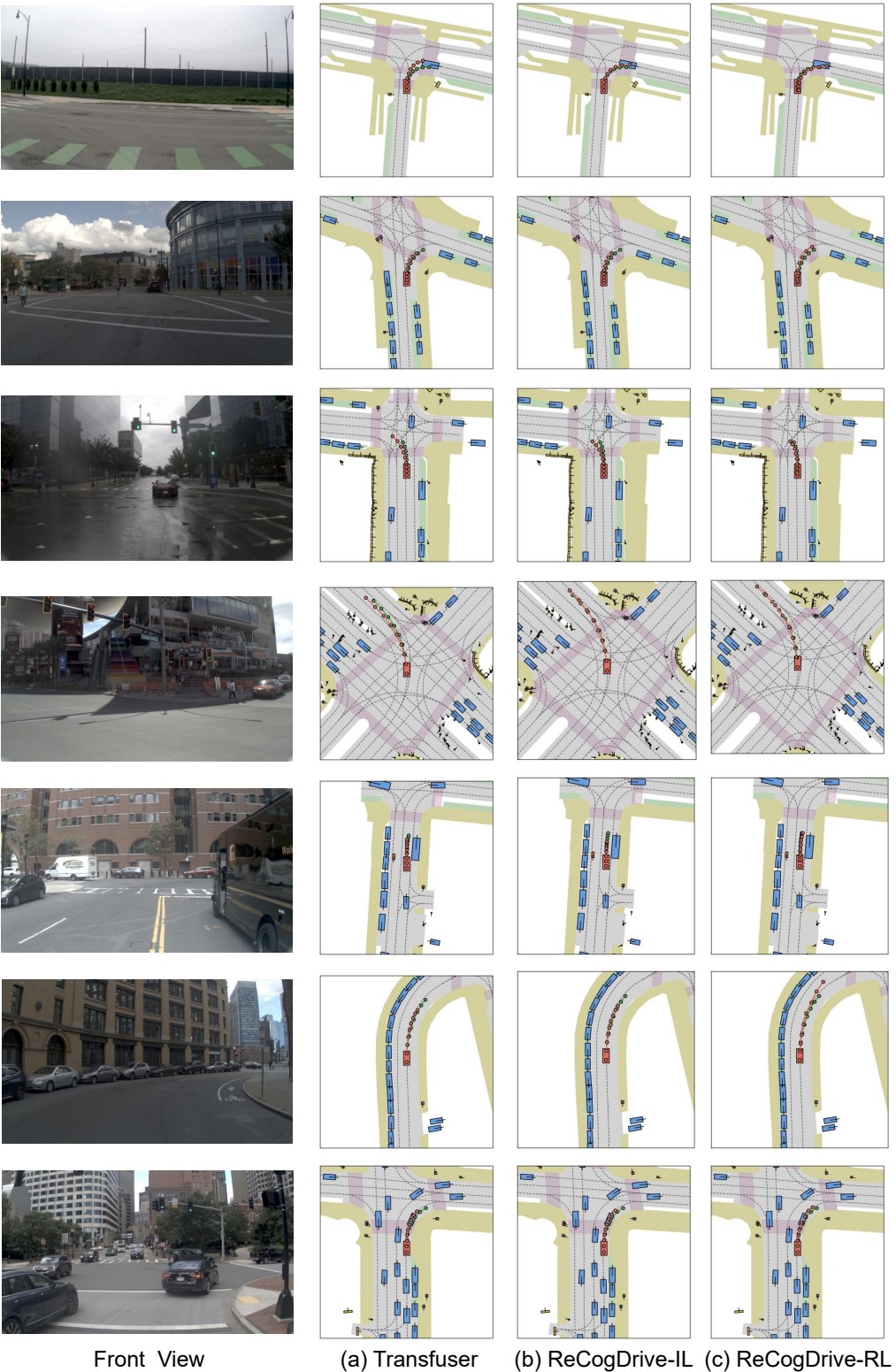

Front View  (a) Transfuser  (b) ReCogDrive-IL  (c) ReCogDrive-RL

Figure 15: Qualitative comparisons on the Navtest benchmark.

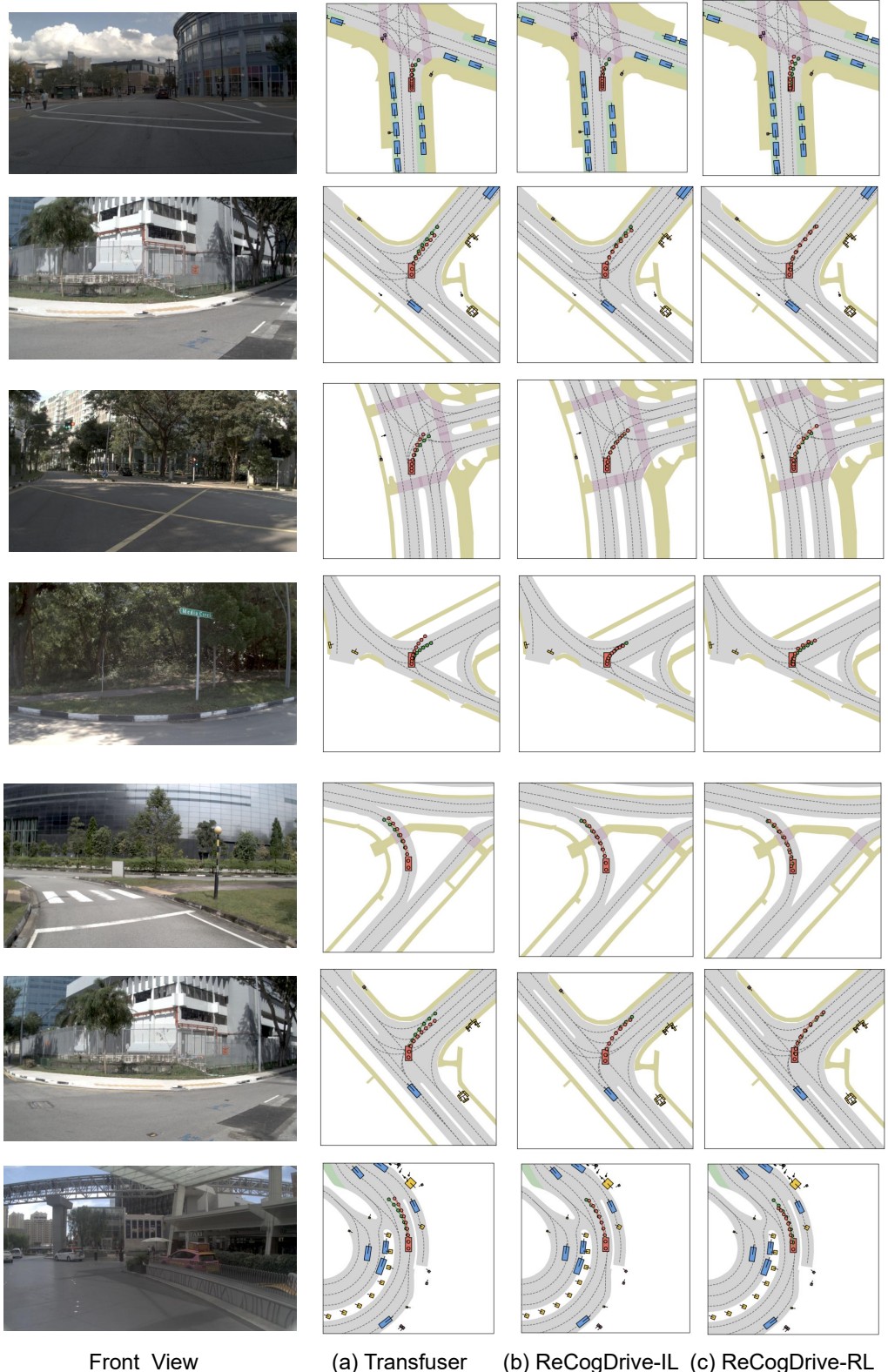

Front View      (a) Transfuser     (b) ReCogDrive-IL   (c) ReCogDrive-RL

Figure 16: Qualitative comparisons on the Navtest benchmark.

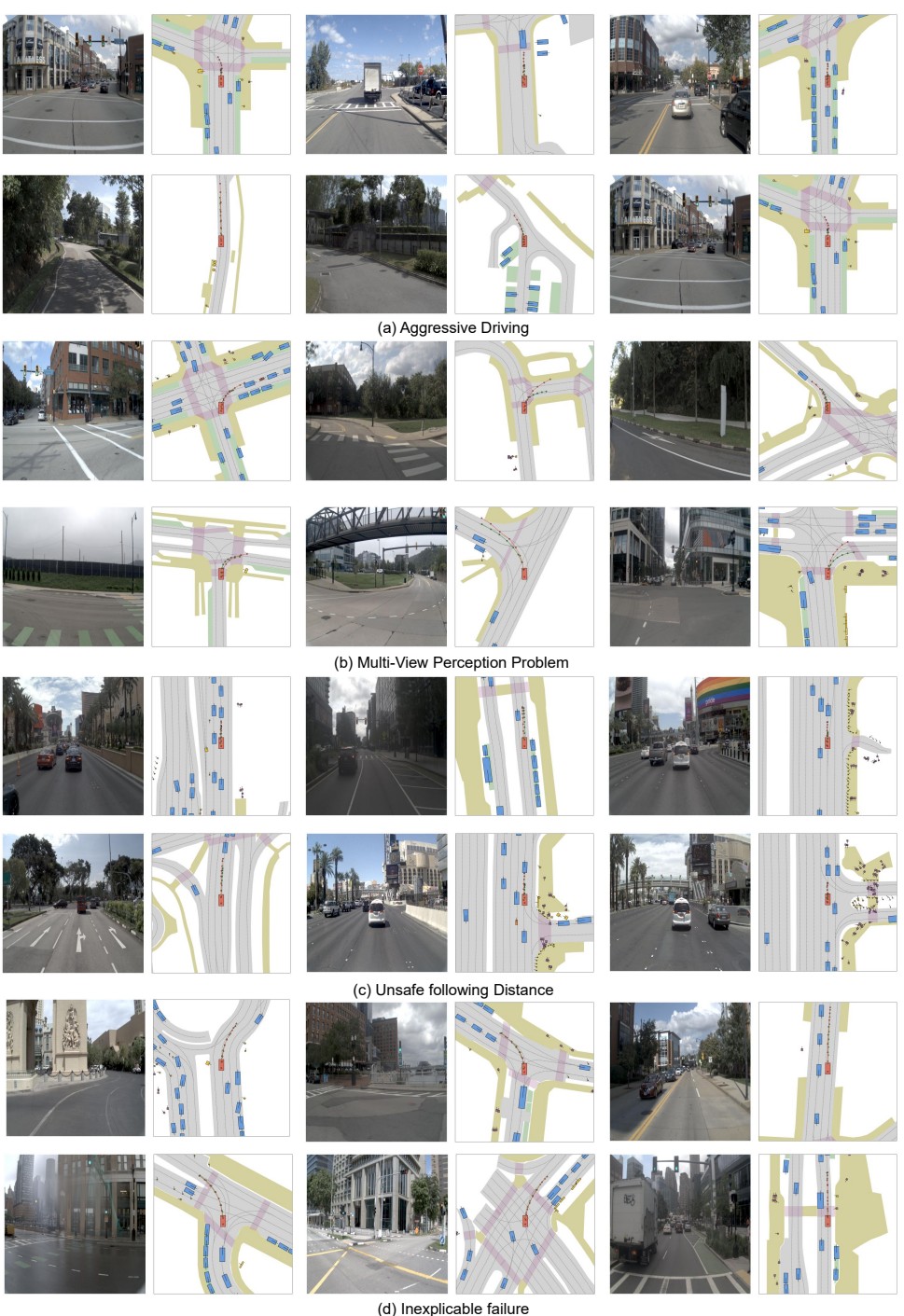

Figure 17: Representative failure cases of ReCogDrive on the Navsim benchmark, including aggressive driving, multi-view perception problem, unsafe following distances, and inexplicable failures. Zoom in for better view.

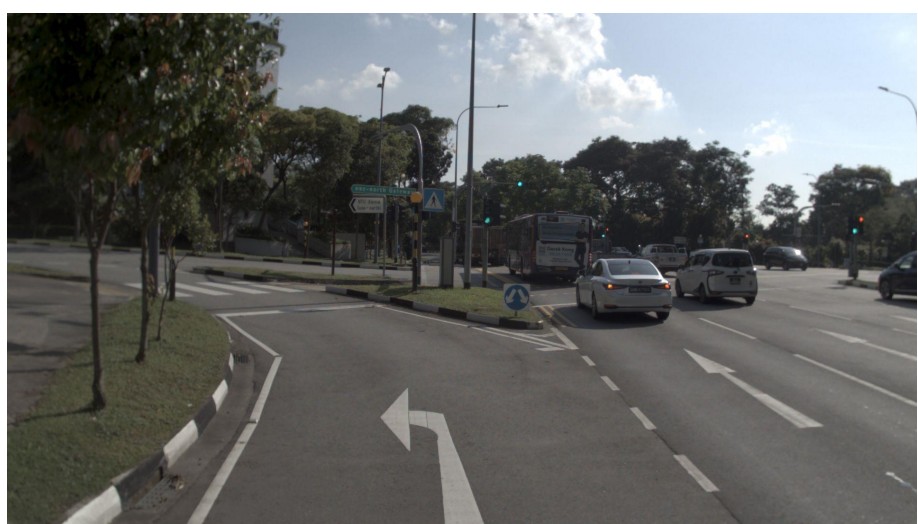

Please make a description for the driving scene.

The scene is set during daylight under a clear sky and scattered clouds. The road is smooth and appears to be a multi-lane highway with no visible obstacles. Traffic lights ahead are showing green, allowing for continued forward movement. A silver sedan is positioned in front on the right lane, moving steadily ahead, while a white van is further up, also proceeding forward. On the far right, a black car is visible, maintaining its lane. A truck is observed slightly ahead on the left, and a bus is further back, both maintaining their lanes and moving forward.

Identify and describe the road markings and signs.

The image shows displays a clear road scene where white dashed lines indicate lane separations, guiding drivers to stay within their respective lanes, while a white arrow on the road suggests the direction of travel for the lane it's in. A round blue sign with a red slash over a black right-turning arrow signals that right turns are prohibited at the junction ahead, requiring drivers to either proceed straight or turn left if permitted. The presence of a pedestrian crossing is implied by the zebra lines on the left, advising drivers to be cautious and ready to stop if pedestrians are present.

Predict the vehicle's trajectory based on the front-view image, historical trajectory and command.

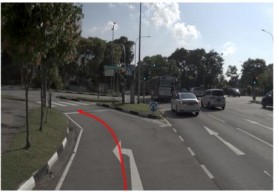

Figure 18: An example shows ReCogDrive's capability in road sign recognition and intersection-turn planning: given a front-view image and a user query, ReCogDrive generates a detailed scene description, identifies key traffic signs at the junction, and predicts a safe turning trajectory.

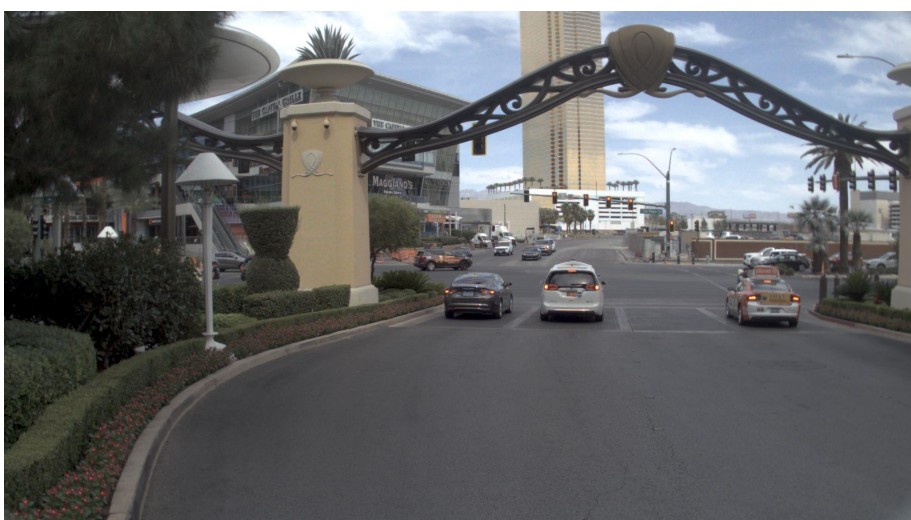

Please make a description for the driving scene.

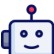 The current driving scene is set during daylight hours under partly cloudy skies, providing ample natural lighting. The road surface appears smooth and clear of any obstacles, with a few vehicles ahead maintaining a steady pace. To the left, there's a line of cars, including a dark sedan and a white sedan approaching an intersection. To the right, a taxi is visible. Caution is advised as the cars ahead are preparing to move through the intersection. The presence of well-maintained roadside greenery and the archway adds to the urban aesthetic but does not impact the driving conditions.

Identify the most influential object that affects the current driving situation.

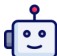 The most influential object in the current driving situation is the traffic light ahead, which is currently red. This traffic light is crucial as it dictates the flow of vehicles and ensures safety at the intersection. The cars ahead, including a silver vehicle directly in front of the viewer, have come to a stop, indicating compliance with the traffic signal. Given the red light, the appropriate driving action is to safely come to a complete stop behind the silver vehicle, maintaining a safe following distance. This will allow the driver to respond appropriately when the light turns green, ensuring a smooth and safe transition through the intersection.

Predict the vehicle's trajectory based on the front-view image, historical trajectory and command.

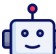 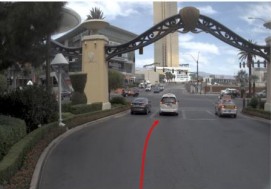

Figure 19: An example shows ReCogDrive's capability in scenario understanding and trajectory planning: at a red-light intersection with braking vehicles ahead, ReCogDrive describes the traffic lights and road conditions, identifies both the traffic light and lead vehicles, and predicts an accurate stop trajectory.

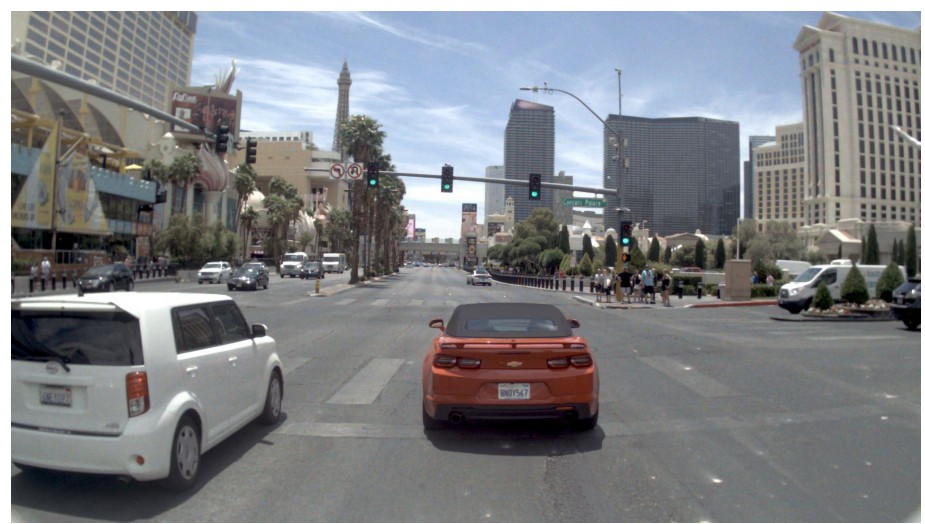

Please make a description for the driving scene.

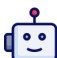 The scene is set during daylight with clear skies, suggesting favorable weather for driving. The road is a multi-lane highway with smooth surfaces. Traffic lights ahead are green, indicating that it's safe to proceed. Several vehicles are present, including a white car to the left and an orange convertible ahead, both moving forward at a moderate speed. Pedestrians can be seen on the sidewalks to the right. The overall setting is calm, with no immediate hazards or congestion. Based on this description, you can proceed with caution, maintaining a safe distance from the vehicles ahead.

Please predict the orange car future driving behaviors

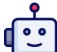 The red car is positioned in the right lane, approaching an intersection with a green traffic light. Based on the visual context, the car will likely continue to keep its current speed and proceed straight through the intersection. This decision is supported by the clear visibility of the green light and the absence of any obstacles or pedestrians in the car's path.

What is your plan for the next three seconds?

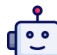 KEEP, STRAIGHT

Predict the vehicle's trajectory based on the front-view image, historical trajectory and command.

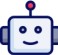 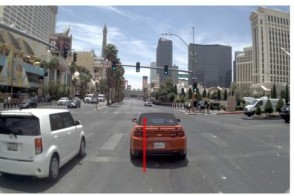

Figure 20: An example shows ReCogDrive's capability in behavior prediction and planning: at a green-light intersection, ReCogDrive describes the road and traffic conditions, identifies the green signal and the lead vehicle's behavior, generates a high-level "KEEP,STRAIGHT" command, and predicts the precise continuous trajectory for the ego vehicle.

