# OpenReview forum: "ReCogDrive: A Reinforced Cognitive Framework for End-to-End Autonomous Driving"
_ICLR.cc/2026/Conference — ICLR 2026 Poster_

### Official Review · Reviewer_wtRP · 2025-10-28

**Soundness:** 2
**Presentation:** 3
**Contribution:** 2
**Rating:** 4
**Confidence:** 5

**Summary:**

ReCogDrive introduces a reinforced cognitive framework that integrates high-level semantic reasoning from a Vision-Language Model (VLM) with continuous control via a Diffusion Planner for end-to-end autonomous driving. Through a three-stage training pipeline—driving Q&A pretraining to bridge the domain gap, imitation learning for language-to-action mapping, and reinforcement learning in simulation for safety and generalization—the model achieves robust, human-like decision-making in rare scenarios.

**Strengths:**

1. Integration of cognition and control: Combines semantic reasoning from a VLM with continuous control from a diffusion planner, forming a full cognitive-to-action pipeline.

2. Three-stage training paradigm: Sequentially applies Q&A pretraining, imitation learning, and reinforcement learning to enhance understanding, controllability, and safety while reducing domain and generalization gaps.

**Weaknesses:**

1.The paper presents a "cognitive framework" that, upon inspection, is a modular integration of existing methods rather than a novel algorithmic solution. The "hierarchical data pipeline" (Sec 4.1) is a data curation strategy, not an algorithmic one. It offers no new mechanism for mitigating domain bias or improving multi-modal alignment. The use of a diffusion planner (Sec 4.2) to connect VLMs to continuous control is not new. This has been demonstrated by recent work (e.g., ORION [1]), making the contribution here incremental at best. The "DiffGRPO" (Sec 4.3) is not a new algorithm but a direct application of the existing GRPO framework to a diffusion planner. Framing this as a key innovation inflates the paper's novelty. Furthermore, this RL stage operates in a decoupled fashion, fine-tuning the action policy without any mechanism to enhance or feed back into the VLM's reasoning process.

2.The central premise of using a VLM is for its reasoning, yet the paper's own results show the VLM is relegated to a simple feature extractor. The paper's own ablation study (Table 3) is damning: the "Driving Pre-training" (the cognitive part) yields a trivial 1.7 PDMS gain. In contrast, the non-cognitive RL fine-tuning provides the main performance boost (4.3 PDMS). Most critically, the appendix (Table 10) confirms that adding Chain-of-Thought (CoT) reasoning provides "no gain" and even hurts performance (a 0.1 PDMS drop). This finding fatally undermines the paper's core "cognitive" premise.

3.The paper's strongest result (+4.3 PDMS) comes from an RL stage that is effectively "training on the test." The model is fine-tuned within the NAVSIM simulator using the benchmark's own evaluation metric (PDMS) as the reward signal. This methodology is highly questionable. This approach raises significant concerns about the fairness of the comparison to other methods and suggests the results are likely overfit to the NAVSIM benchmark, severely limiting any claims of generalizability.

4.The experimental comparison (Table 1) is weak. The paper fails to benchmark against key contemporary VLA-specific models (e.g., AutoVLA [2]), a notable omission in this fast-moving subfield. Instead, the comparisons are against general-purpose E2E models or VLM models. This makes it impossible to assess the paper's true contribution to VLA research.

5.Finally, the experimental setup is ambiguous in several key areas, hindering reproducibility. The exact training data mix is unclear. The paper mentions multiple VQA datasets (Appx. D) plus 775K newly generated pairs, but the final composition and weighting for the reported models are not specified. The Bench2Drive protocol (Table 2) is unexplained. It's impossible to tell if these are zero-shot results or if the model was fine-tuned, making the scores difficult to interpret. The tables themselves are sloppy. Table 3 uses "100" and "100.0" inconsistently in the same column. Table 4 introduces an "Err. (%)" metric without a clear definition. The small differences are highlighted in red, creating unnecessary confusion.

[1] ORION: A Holistic End-to-End Autonomous Driving Framework. ICCV 2025

[2] AutoVLA: A Vision-Language-Action Model for End-to-End Autonomous Driving with Adaptive Reasoning and Reinforcement Fine-Tuning. NeurIPS 2025

**Questions:**

see weakness

---

> ### Author Response · Authors · 2025-11-22
> **Response to Reviewer wtRP 1/3**
>
> We thank the reviewer for the thoughtful and constructive feedback. Our responses to each point are provided below.
>
> **1. Clarifying Contributions and Novelty.** ReCogDrive is a reinforced cognitive driving framework tackling three key VLA limitations: missing driving priors in VLMs, unreliable text-based trajectory, and multimodal collapse under imitation learning.
>
> **(1) Regarding the hierarchical data pipeline.** Our hierarchical driving data pipeline is not simple data curation. It is designed to generate large-scale, high-quality cognitive data across four levels: perception, dynamic understanding, planning and reasoning, and advanced reasoning. It covers a far richer task set than prior driving-QA datasets, including traffic-flow analysis, key-agent detection, scene assessment, counterfactual reasoning, and 3D grounding. This broader supervision injects driving-specific priors that directly benefit the planner.
>
>
> Moreover, we show empirically that **driving pre-training is key to injecting driving priors**. Across Bench2Drive (DS, RC) and NAVSIM (PDMS), it delivers substantial and consistent gains:
>
> | ReCogDrive-VLM-8B | DS | RC | PDMS |
> |-|-|-|-|
> | w/o driving pre-train | 43.2 | 14.1 | 83.3 |
> | w/ driving pre-train | 62.1 | 31.7 | 86.4 |
>
>
> These results reinforce the value of driving pre-training. It also surpasses CoT-style SFT: AutoVLA-SFT reaches 80.5 PDMS (3B), while our 2B ReCogDrive-VLM reaches 84.1 PDMS with fewer parameters and a single front view, **indicating that driving pre-training provides stronger cognition than generic CoT tuning**.
>
> **Our hierarchical pipeline is also scalable.** Larger and higher-quality corpora consistently improve PDMS, indicating that richer cognitive signals directly enhance the planner.
>
> | Data | PDMS |
> |-|-|
> | 85K (Only Traj) | 83.3 |
> | 1.5M (Low Quality) | 84.6 |
> | 3.2M (Low Quality) | 85.3 |
> | 3.1M (High Quality) | 86.4 |
>
> **This consistent improvement shows that the hierarchical pipeline provides meaningful, scalable driving cognition rather than simple data curation.**
>
>
> **(2) Regarding the comparison with ORION.** ORION uses a VAE decoder within QT-Former and compares it to a VLM paired with DiffusionDrive, but it includes neither diffusion-specific design nor any RL fine-tuning. In contrast, ReCogDrive conditions diffusion on full VLM hidden states, injects ego-history information through AdaLN, and optimizes the denoising process with DiffGRPO. For a fair comparison, we also build a VLM+VAE baseline using the same VLM conditioning and compare it directly with our cognition-guided diffusion planner:
>
> | Decoder  | PDMS |
> |-|-|
> | VAE Decoder | 83.0 |
> | Diffusion Planner  | 86.5 |
> | VAE Decoder + Top-k | 83.0 |
> | Diffusion Planner + Top-k | 89.3 |
>
> Under identical VLM conditioning, our diffusion planner achieves higher PDMS than the VAE decoder (86.5 vs 83.0). The VAE decoder also suffers from two fundamental limitations:
>
> 1. **No multi-modality.** It collapses futures into an averaged trajectory, while our diffusion head produces diverse samples, reaching 89.3 PDMS under Top-k (k=10) compared with 83.0 for the VAE.
>
> 2. **Incompatible with RL.** It provides no tractable trajectory likelihood, preventing policy-gradient optimization, whereas diffusion supports GRPO through explicit denoising likelihoods.
>
> **(3) Regarding the comment that DiffGRPO is a direct application of the existing GRPO framework.** Existing GRPO formulations are designed for LLM and do not provide a framework for diffusion planners. Our contribution is to innovatively extend GRPO to a diffusion planner within a VLA for autonomous driving, which has not been explored before. Unlike methods such as AlphaDrive that apply GRPO to VLM, DiffGRPO maps the diffusion denoising process into a multi-step MDP, leverages diffusion’s multimodal trajectory sampling, and uses a PDMS reward to optimize safe, feasible trajectories with RL.
>
> Since submission, several concurrent works have adopted similar GRPO-style objectives. Their NAVSIM PDMS scores are:
>
> | Method | PDMS |
> |-|-|
> | TrajHF  | 87.6 |
> | CoIRL-AD | 88.2 |
> | DIVER  | 88.3 |
> | InternVL +  GRPO | 88.3 |
> | AutoVLA   | 89.1 |
> | AlphaDrive  | 89.5 |
> | DriveDPO  | 90.0 |
> | ReCogDrive | 90.8 |
>
> ReCogDrive achieves the highest PDMS despite using only a 2B VLM and a single front-view input, demonstrating the effectiveness of our DiffGRPO formulation.
>
> **Regarding the concern that RL is “decoupled” from cognition**, we freeze the VLM during RL to preserve its reasoning ability and prevent the cognitive priors from collapsing under a different training objective. DiffGRPO optimizes a diffusion policy that conditions on the VLM’s hidden states at every denoising step, so the planner continues to rely on these priors throughout RL. Empirically, applying GRPO to a trajectory-only InternVL backbone yields 88.3 PDMS, while our full model reaches 90.8 PDMS (+2.5), confirming that RL benefits from the cognitive priors established during driving pre-training.

---

> ### Author Response · Authors · 2025-11-22
> **Response to Reviewer wtRP 2/3**
>
> **2. Clarification on the Cognition Component in Our Framework**
>
>
> Thank you for raising this important point. As discussed in the appendix (Q2), our design follows how humans acquire driving skills: after sufficient practice, humans internalize driving cognitive priors and can drive stably without explicit reasoning, relying on deeper cognition only in rare or complex situations. Similarly, large-scale driving pre-training equips the VLM with driving-specific cognitive priors that allow it to handle the vast majority of scenarios, while optional reasoning is only needed for a small set of challenging cases.
>
> **(1) Regarding the 1.7 PDMS gain from Driving Pre-training, we clarify three points.**
>
> - **The limited gain at 2B mainly reflects insufficient model capacity.** At small scale, the model cannot fully absorb driving cognition, resulting in a 1.7 PDMS improvement. When scaling to 8B, the effect becomes much stronger:
>
>    | Method               |  NC  | DAC  | TTC  | C. |  EP  | PDMS |
>    |-|-|-|-|-|-|-|
>    | InternVL-Traj (2B)   | 97.4 | 91.3 | 93.0 | 100   | 77.2 | 82.4 |
>    | ReCogDrive-VLM (2B)  | 97.6 | 93.1 | 92.7 | 100   | 79.1 | 84.1 |
>    | InternVL-Traj (8B)   | 97.0 | 92.4 | 91.8 | 100   | 78.9 | 83.3 |
>    | ReCogDrive-VLM (8B)  | 98.2 | 94.7 | 94.6 | 100   | 80.6 | 86.4 |
>
>    Driving pre-training brings a **3.1 PDMS** gain at 8B, demonstrating that cognition becomes substantially more effective with adequate model capacity.
>
> - **Even at 2B scale, the gain from driving QA pre-training is meaningful relative to other VLA-style methods.** Compared to AutoVLA-SFT (3B), which relies on CoT-style SFT, our 2B cognitively trained model achieves substantially higher PDMS despite using fewer parameters and only a single front view:
>
>    | Method             |  NC  | DAC  | TTC  | C. |  EP  | PDMS |
>    |-|-|-|-|-|-|-|
>    | AutoVLA-SFT (3B)   | 96.9 | 92.4 | 88.1 | 99.9 | 75.8 | 80.5 |
>    | ReCogDrive-VLM (2B)| 97.6 | 93.1 | 92.7 | 100  | 79.1 | 84.1 |
>
>    Despite using **fewer parameters and only a single front view**, ReCogDrive-VLM (2B) still outperforms AutoVLA-SFT (3B) by **+3.6 PDMS**. This indicates that driving-oriented pre-training is more impactful than standard CoT-style SFT, and it would be misleading to assess cognition solely by comparing its gain to that of the RL stage, which optimizes a different objective on top of a cognitively trained policy.
>
>
> - **Driving pre-training improves closed-loop driving performance beyond NAVSIM.** Bench2Drive further validates the benefit of the cognitive stage:
>
>    | Method            |  DS   |  RC  |
>    |-|-|-|
>    | InternVL-FT-Traj  | 43.2 | 14.1 |
>    | ReCogDrive-VLM    | 62.1 | 31.7 |
>
>    Driving pre-training yields substantial improvements in both Driving Score (DS) and Route Completion (RC), two closed-loop metrics closely tied to real driving quality. This further shows that the cognitive stage meaningfully strengthens the VLM’s driving ability, and that RL should be viewed as a complementary refinement step that builds on an already cognitively informed policy, rather than a substitute for cognition.
>
>
>
> **(2) Regarding the claim that RL contributes the main gain and thus overshadows cognition**, we note that RL itself benefits from cognition. Comparing GRPO applied to a trajectory-only VLM vs. our full cognitively trained model:
>
>
> | Method                                    |  NC  | DAC  | TTC  | C. |  EP  | PDMS |
> |-|-|-|-|-|-|-|
> | InternVL (w/o driving pre-train) + GRPO   | 98.2 | 96.1 | 94.3 | 100   | 83.5 | 88.3 |
> | ReCogDrive (w/ driving pre-train)         | 97.9 | 97.3 | 94.9 | 100   | 87.3 | 90.8 |
>
> ReCogDrive improves PDMS by **+2.5** over InternVL + GRPO, indicating that the RL fine-tuning stage is not purely non-cognitive. Instead, driving QA pre-training provides useful cognition that further enhances the effectiveness of GRPO.
>
>
>
> **(3) Regarding the observation that naive CoT yields no improvement**, this is primarily because the policy after DiffGRPO is already close to saturation, leaving limited headroom for additional gains from naive CoT. We also evaluate the VLM as a high-level driving guide for non-VLM architectures and for the VLA policy.
>
> | Method                      |  NC  | DAC  | TTC  | C. |  EP  | PDMS |
> |-|-|-|-|-|-|-|
> | Transfuser                  | 97.7 | 92.8 | 92.8 | 100   | 79.2 | 84.0 |
> | Transfuser-ReCogDrive       | 98.0 | 95.1 | 94.2 | 100   | 81.2 | 86.7 |
> | DiffusionDrive              | 98.2 | 96.2 | 94.8 | 100   | 82.2 | 88.1 |
> | DiffusionDrive-ReCogDrive   | 98.9 | 96.4 | 95.9 | 100   | 83.0 | 89.0 |
> | ReCogDrive-IL             | 98.1 | 94.7 | 94.2 | 100  | 80.9 | 86.5 |
> | ReCogDrive-IL+High-Level Command | 98.6 | 94.5 | 95.4 | 100  | 81.4 | 87.1 |
>
> These consistent PDMS gains show that the VLM provides genuine cognitive guidance across planners, rather than merely supplying features. This supports our cognition-guided design and demonstrates that cognition contributes beyond explicit CoT generation.

---

> ### Author Response · Authors · 2025-11-22
> **Response to Reviewer wtRP 3/3**
>
> **3. Clarification on the RL Fine-Tuning Procedure.**
> First, our RL stage is conducted **only on the Navtrain split**, and all evaluations are performed exclusively on Navtest, so the model is **not trained on the test set**.
>
> Second, PDMS is not a dataset-specific annotation; it evaluates whether a predicted trajectory results in collisions, comfort violations, or out-of-lane driving. **These are generic driving-safety signals independent of any specific test trajectory labels.** This makes PDMS a natural and principled reward for improving the quality of predicted trajectories through RL.
>
> Moreover, several prior works on NAVSIM adopt similar practices, incorporating PDMS into their training pipelines for filtering, selection, or distillation, such as Hydra-MDP, GoalFlow (CVPR 2025), WOTE (ICCV 2025), DriveDPO (NeurIPS 2025) and AutoVLA (NeurIPS 2025), demonstrating that **using PDMS as a reward in training is a widely accepted and validated approach.**
>
> To verify generalization beyond NAVSIM, we also evaluate on Bench2Drive. ReCogDrive-RL again improves over ReCogDrive-IL:
>
> | Method            | DS   | RC   |
> |:-|:-:|:-:|
> | ReCogDrive-IL     | 64.5 | 20.0 |
> | ReCogDrive-RL     | 85.4 | 60.0 |
>
> Following DriveTransformer (ICLR 2025), we use the Dev10 validation subset for efficient evaluation. These results confirm that **the RL stage does not overfit NAVSIM and yields consistent gains on another benchmark.**
>
>
> **4. Comparison with AutoVLA.** AutoVLA was released after our work and is essentially concurrent. On NAVSIM there were no public VLA models available, so we originally compared against strong generic VLMs (QwenVL3, InternVL3). For completeness, we now include comparisons with AutoVLA on NAVSIM:
>
> | Method         | Params | Sensors | NC | DAC | TTC | Conf | EP | PDMS |
> |:-|:-:|:-:|:-:|:-:|:-:|:-:|:-:|:-:|
> | AutoVLA-IL     | 3B     | Multi   | 96.9 | 92.4 | 88.1 | 99.9 | 75.8 | 80.5 |
> | ReCogDrive-IL  | 2B     | Front   | 98.1 | 94.7 | 94.2 | 100  | 80.9 | **86.5** |
> | AutoVLA-RL     | 3B     | Multi   | 98.4 | 95.6 | 98.0 | 99.9 | 81.9 | 89.1 |
> | ReCogDrive-RL  | 2B     | Front   | 97.9 | 97.3 | 94.9 | 100  | 87.3 | **90.8** |
> | AutoVLA-Topk       | 3B | Multi  | 99.1 | 97.1 | 97.1 | 100 | 87.6 | 92.1 |
> | ReCogDrive-Topk    | 2B | Front  | 98.7 | 98.5 | 96.6 | 100 | 88.2 | **92.6** |
>
>
> Overall, **ReCogDrive (2B, front-view) achieves higher PDMS scores than AutoVLA (3B, multi-view)**, demonstrating that our cognition-guided diffusion plus RL design yields a more effective policy with fewer parameters and views.
>
>
> For completeness, we also evaluate the Top-k protocol used in AutoVLA, where the model generates multiple candidate trajectories and reports the highest PDMS among them. **ReCogDrive also performs better under this setting**, though we view Top-k as less representative of realistic single-trajectory deployment and therefore omit it from the main paper.
>
> | Method         | Inference Time (s/frame) |
> |:-|:-:|
> | AutoVLA-Slow       | 10.51                        |
> | AutoVLA-Fast       | 1.07                        |
> | ReCogDrive    | 0.08                        |
>
> Moreover, **ReCogDrive also runs significantly faster than AutoVLA**, reducing per-frame latency by more than an order of magnitude.
>
> **We have updated Tab. 1 in the revised version to include AutoVLA and also added the appropriate citation.**
>
>
> **5. Experimental Setup and Reporting Details.** Regarding reproducibility, we confirm that all code, datasets, pretrained models, and full pipelines will be publicly released.
>
> **Training data composition.** On NAVSIM, the model is trained using NAVSIM trajectory data, our automatically generated 775K QA pairs, 2.3M open-source driving QA samples, and 665K LLaVA-style instruction data, with respective sampling proportions of 5%, 22.9%, 68.1%, and 3.9%. We then fine-tune the NAVSIM-trained model on Bench2Drive using Bench2Drive trajectory data and Bench2Drive QA data, sampled at 79.2% and 20.8%. **We have added these details to Appendix D.3 (Dataset Statistics) and Appendix E (Training Configuration) in the supplementary material.**
>
> **Reporting clarity.** We have now unified all numeric formats across the paper. For Tab. 4, **we have added a clear definition of the “Err. (%)” metric** in the Comparison of Trajectory Generation Methods section: it measures the probability that a model produces an unparsable trajectory. This issue arises only in plain-text trajectory generators such as EMMA and OmniDrive, where autoregressive decoding may occasionally produce outputs that do not match the trajectory template. Although the error rate is very small (as low as 0.01%), **even such rare failures are unacceptable for safety-critical autonomous driving**. We therefore report this metric to highlight the practical limitations of plain-text trajectory generation compared with end-to-end or diffusion-based approaches, which do not suffer from this issue.

---

### Official Review · Reviewer_diW3 · 2025-10-31

**Soundness:** 3
**Presentation:** 4
**Contribution:** 3
**Rating:** 6
**Confidence:** 5

**Summary:**

This paper proposes ReCogDrive, a novel end-to-end autonomous driving framework that integrates a Vision-Language Model (VLM) for cognitive reasoning with a diffusion planner to generate continuous trajectories. The framework is further refined using a reinforcement learning stage called DiffGRPO to improve safety and comfort, achieving state-of-the-art results on the NAVSIM and Bench2Drive benchmarks.

**Strengths:**

1. Well-designed Framework: The core architecture that integrates a cognitive VLM, a diffusion planner, and an RL refinement stage is well designed. It provides an effective solution to the modality mismatch problem for VLM-based agents.
2. Strong Empirical Validation: The paper's claims are well-supported by strong empirical performance on two challenging benchmarks (NAVSIM and Bench2Drive), comprehensive ablation studies that isolate the gains from each component, and a methodical data curation pipeline.

**Weaknesses:**

**Major Weaknesses:**

1. Lack of Motivation for GRPO: The paper provides insufficient insight into the choice of Group Relative Policy Optimization (GRPO). It is not clear why GRPO is better suited for optimizing a diffusion policy in this context compared to other well-established RL algorithms (e.g., PPO, or a simpler policy gradient method like REINFORCE).
2. Unclear DiffGRPO Algorithm Design: The description of the DiffGRPO algorithm seems to imply a fundamental difference from typical GRPO. In GRPO, trajectories within one group are computed starting from the same initial state. This design choice is critical for the advantage calculation to be valid, and the paper needs to elaborate more on this.

**Minor Weaknesses:**

1. Unclear Methodology and Data Pipeline: The methodology, particularly regarding the specifics of the data pipeline (Section 4.1) and the diffusion planner architecture (Section 4.2), is explained at a high level and lacks detailed examples for full clarity.
2. Messy Symbol Use: The paper suffers from unclear and at times inconsistent mathematical notation (e.g., $E_{act}$, $E_{hist}$, $D_{act}$, $x_{hist}$ are undefined, and $r_{1...G}$ in Eq. 9 is ambiguous).

**Questions:**

1. Could you please provide a more precise definition of the problem setup? Specifically, what are the exact of the inputs $I_{cam}$, $L_{nav}$, and $S_{ego}$? Similarly, how is the output $V_{traj}$ defined? What is the rationale for predicting headings when waypoints (which imply direction) are already being generated?
2. In Section 4.2, what is the architectural motivation for using both the full VLM hidden states $F_h$ and the pooled global embedding $\bar{F}_h$? What distinct roles do these two forms of conditioning play?

---

> ### Author Response · Authors · 2025-11-22
> **Response to Reviewer diW3 1/2**
>
> We thank the reviewer for the thoughtful evaluation and constructive feedback. We appreciate the positive assessment and address the comments below.
>
> **1. Comparison with Other RL Algorithms.** We clarify our motivation here. In autonomous driving, as illustrated in Fig. 4 of the main paper, a single scene typically admits multiple safe trajectories rather than a unique ground-truth path. Our DiffGRPO samples a group of trajectories under the same scene condition and computes a group-relative advantage, so that the policy is explicitly encouraged to increase the likelihood of safer and more stable rollouts. This group-wise computation naturally matches the multi-ground-truth nature of autonomous driving scenarios.
>
> **Compared to standard REINFORCE**, GRPO computes a group-wise baseline within each scene, decoupling trajectory-quality comparisons from cross-scene difficulty. This group computation is better aligned with the multi-modal diffusion planner, as it directly ranks trajectories against other candidates in the same scenario rather than against an averaged signal across heterogeneous scenes.
>
>
> **Compared to PPO**, using PPO would require an additional critic of similar size to the diffusion planner, significantly increasing compute and memory. In highly multi-modal settings with sparse, delayed rewards, the critic also struggles to yield stable, low-variance advantages. GRPO avoids value-function learning and instead uses simple group computation for effective variance reduction.
>
> We additionally implement REINFORCE, DPPO (Diffusion Policy Policy Optimization; a PPO-style RL method for diffusion policies), and DiffGRPO on the same diffusion planner and PDMS reward:
>
> | Algorithm  | NC   | DAC  | TTC  | C. | EP   | PDMS |
> |:-:|:-:|:-:|:-:|:-:|:-:|:-:|
> | REINFORCE  | 98.5 | 97.3 | 95.9 | 100  | 82.9 | 89.5 |
> | DPPO        | 98.6 | 97.6 | 96.0 | 100  | 82.4 | 89.5 |
> | **DiffGRPO** | 97.9 | 97.3 | 94.9 | 100  | 87.3 | **90.8** |
>
> **DiffGRPO achieves the best PDMS score, indicating superior overall driving performance. We have added comparisons in Tab.5 and Fig.5 of the main paper.**
>
> **2. About DiffGRPO Algorithm Design.**  We clarify the formulation. The denoising process of the diffusion planner can be viewed as a multi-step MDP:
>
> $$
> \begin{aligned}
> s_t &\\triangleq (c, t, x_t)
> &\\qquad
> \\pi_\\theta(a_t \mid s_t) &\\triangleq p_\\theta(x_{t-1}\mid x_t, t, c)
> &
> P(s_{t+1}\mid s_t, a_t) &\\triangleq (\\delta_c,\\delta_{t-1},\\delta_{x_{t-1}})
> \\\\[6pt]
> a_t &\\triangleq x_{t-1}
> &\\qquad
> \\rho_0(s_0) &\\triangleq (c,\\delta_T, \\mathcal{N}(0,I))
> &
> R(s_t,a_t) &\\triangleq
> \\begin{cases}
> r(x_0, c), & t=0\\\\
> 0, & \\text{otherwise}
> \\end{cases}
> \\end{aligned}
> $$
>
> In this formulation, each MDP state is defined as $s_t = (c, t, x_t)$, where $c$ collects all conditioning inputs to the diffusion planner (VLM final hidden states, ego status, and historical trajectory) and $x_t$ is the noisy trajectory at timestep $t$; the action is $a_t = x_{t-1}$, corresponding to the model-predicted denoised trajectory for the next timestep. The policy term $\\pi_\\theta(a_t \mid s_t) = p_\\theta(x_{t-1}\mid x_t, t, c)$ is the Gaussian denoising distribution produced by the diffusion model, while the transition kernel $P(s_{t+1}\mid s_t, a_t)$ is deterministic and simply updates the state by keeping $c$ fixed, decreasing the timestep from $t$ to $t-1$, and setting the next noisy state to the sampled action $x_{t-1}$. Finally, the reward function $R(s_t,a_t)$ assigns the PDMS score $r(x_0,c)$ only at the final denoising step (and zero elsewhere), so that the entire diffusion chain is treated as a single composite action for credit assignment.
>
> The initial-state distribution $\\rho_0(s_0) = (c, \\delta_T, \\mathcal{N}(0,I))$ means that all trajectories in a group share the same conditioning $c$, start from the same denoising horizon $T$, and draw $x_T$ from the same Gaussian prior $x_T \sim \\mathcal{N}(0,I)$. **This is consistent with standard GRPO, where trajectories within one group are computed starting from the same initial state.**
>
>
> **3. About Methodology and Data Pipeline.**
>
> **Regarding Section 4.1 Data Pipeline**, the main paper primarily presents the scalable hierarchical data pipeline with its multi-level design and multi-stage processing overview. We have **substantially expanded Supplementary Section D** with full details, including dataset sources, the three-stage procedure (annotation, normalization, scoring/filtering), as well as the training data distribution statistics. In addition, **we have added a complete QA example in Fig. 7 and will release all pipeline scripts, processing code, and the finalized QA data.**
>
> **Regarding Section 4.2 Diffusion Planner**, we have updated the main paper to **add details and explicitly define all symbols** used in the diffusion planner. In addition, we provide **a detailed specification of the diffusion planner architecture and hyperparameters** in **supplementary Section E**.

---

> ### Author Response · Authors · 2025-11-22
> **Response to Reviewer diW3 2/2**
>
> **4. Definition of Symbols in the Paper.** Thank you for pointing this out. We have clarified all previously ambiguous symbols in the revised manuscript. Specifically, $E\_{\text{act}}$ and $E\_{\text{his}}$ denote the action encoder and history–trajectory encoder, $D\_{\text{act}}$ is the action decoder used to reconstruct the denoised trajectory, and $x\_{\text{hist}}$ corresponds to the historical ego trajectory $I\_{\text{hist}}$. The fused latent is
>
> $$
> z\_t = \text{concat}\left( E\_{\mathrm{act}}(\mathbf{x}\_t),\ E\_{\mathrm{his}}(I\_{\mathrm{hist}}),\ \bar{F}\_h \right)
> $$
>
> and the denoising step is
>
> $$
> \mathbf{x}\_{t-1}
> = D\_{\mathrm{act}}\Bigl( \text{DiT}\_\theta(z\_t;\ F\_h;\ S\_{\mathrm{ego}};\ t) \Bigr).
> $$
>
> We also replace the ambiguous notation \(r\_{1..G}\) with explicit definitions:
>
> $$
> \hat{A}\_i = \frac{r\_i - \mu}{\sigma},\qquad
> \mu = \frac{1}{G}\sum\_{j=1}^G r\_j,\qquad
> \sigma^2 = \frac{1}{G}\sum\_{j=1}^G ( r\_j - \mu )^2.
> $$
>
> **All of these definitions and notational fixes have been added to the main paper (Section 4.2) to ensure clarity and consistency.**
>
> **5. Definition of the problem setup.**
> Thank you for pointing this out. We have clarified the problem setup in the revised manuscript. Concretely, the inputs are now defined as follows:
>
> - $I\_{\text{cam}} \in \mathbb{R}^{K \times V \times H \times W \times 3}$: a stack of $K$ RGB frames from $V$ cameras, including the current and historical images.
> - $I\_{\text{hist}} \in \mathbb{R}^{T\_{\text{hist}} \times 3}$: the historical ego trajectory, represented as past waypoints and headings.
> - $S\_{\text{ego}}$: the current ego state, including ego velocity $(v\_x, v\_y) \in \mathbb{R}^2$ and ego acceleration $(a\_x, a\_y) \in \mathbb{R}^2$.
>
> The model outputs a future trajectory
>
> $$
> V\_{\text{traj}} = \{(x\_t,\ y\_t,\ \theta\_t)\}\_{t=1}^T \in \mathbb{R}^{T \times 3},
> $$
>
> where $(x\_t, y\_t)$ are the predicted 2D waypoints and $\theta\_t$ are the corresponding heading angles.
>
> Regarding the rationale for predicting headings in addition to waypoints: although finite differences of $(x\_t, y\_t)$ can, in principle, imply a direction, explicit headings are required for downstream closed-loop evaluation and control. In particular, $\theta\_t$ is used to construct oriented ego polygons for collision and TTC checks, to assess driving-direction compliance (e.g., oncoming-traffic violations), and to compute yaw-related comfort metrics (yaw rate and yaw acceleration). Predicting $\theta\_t$ directly is also more numerically stable than recovering it from sparsely sampled waypoints.
>
> **These definitions and the motivation for including headings have been incorporated into the Problem Definition paragraph in the revised main paper.**
>
>
>
> **6. Motivation for $F\_h$ and $\bar{F}\_h$.**
> Thank you for pointing this out. The full VLM hidden states $F\_h$ contain fine-grained latent features that encode the model’s understanding of both the current instruction and the visual context, capturing the detailed intent of the reasoning process. In contrast, the pooled global embedding $\bar{F}\_h$ provides a compact summary of the overall scene and instruction. When concatenated with the noised action and history embeddings, this global representation stabilizes trajectory prediction and improves planning consistency.
>
> We provide an ablation on whether to include the pooled global embedding on top of $F\_h$ under IL-only training, as shown below (note that $F\_h$  is used in both rows):
>
>
>
> | $\bar{F}\_h$ |  NC  | DAC  | TTC  | C. |  EP  | PDMS |
> |:-:|:-:|:-:|:-:|:-:|:-:|:-:|
> | ✗             | 98.1 | 94.2 | 94.0 | 100  | 80.5 | 85.9 |
> | ✓             | 98.1 | 94.7 | 94.2 | 100  | 80.9 | 86.5 |
>
> In addition, we also compare two different ways of constructing the token-level conditioning $F\_h$ itself:
>
> | $F\_h$                |  NC  | DAC  | TTC  | C. |  EP  | PDMS |
> |:-|:-:|:-:|:-:|:-:|:-:|:-:|
> | All Last Hidden State | 98.1 | 94.7 | 94.2 | 100  | 80.9 | 86.5 |
> | Learnable queries + Cross-Attention | 98.0 | 93.9 | 93.9 | 100  | 80.0 | 85.5 |
>
> Here, **All Last Hidden State** feeds the complete VLM hidden tokens into the planner as $F\_h$, whereas **Learnable Queries + Cross-Attention** replaces them with a compressed representation obtained through cross-attention. The results show that using the full hidden states provides stronger, more informative conditioning and yields consistently better planning performance.

---

### Official Review · Reviewer_gJsD · 2025-11-01

**Soundness:** 2
**Presentation:** 2
**Contribution:** 2
**Rating:** 6
**Confidence:** 4

**Summary:**

The paper proposes ReCogDrive, to address challenges in end-to-end autonomous driving framework that integrates a Vision-Language Model (VLM) with a diffusion-based planner and a reinforcement learning (RL) module.
The key contributions include:
Integrates the diffusion planner with VLM models that translates VLM outputs into continuous, stable driving trajectories, addressing the modality mismatch between language and action spaces in way to provide cognitive guidance and enabling precise trajectory generation.
A Diffusion Group Relative Policy Optimization (DiffGRPO) algorithm that fine-tunes the planner using RL to improve safety and comfort.

The model is evaluated on NAVSIM and Bench2Drive benchmarks, achieving state-of-the-art performance in both open-loop and closed-loop settings. It also demonstrates superior performance on DriveLM and DriveBench for visual question answering (VQA) and reasoning tasks.

**Strengths:**

- The integration of VLMs with diffusion models and reinforcement learning addresses key limitations in current end-to-end driving systems.
- The structured approach to data generation and refinement is scalable, and enables the creation of high-quality VQA datasets for autonomous driving.
- The use of a diffusion planner to bridge the gap between discrete language outputs and continuous control actions.
- The introduction of DiffGRPO is a thoughtful addition that enhances the planner’s ability to generate safer and more comfortable trajectories.
- The paper presents extensive experiments, ablation studies, and qualitative analyses across multiple benchmarks, demonstrating the robustness and generalization of the approach.

**Weaknesses:**

- Diffusion Planner: integration of diffusion planner with VLM is explored in ORION. Despite improvements using  the diffusion-based approach still incurs higher inference latency compared to VLP, DiMA which distills VLM knowledge to simpler planners. Comparisons with these relevant methods like ORION, VLP and DiMA are missing.
- Zero shot testing: The model is evaluated in simulation environments (NAVSIM, CARLA), but lacks real-world deployment or testing, which is crucial for autonomous driving applications. It would be if authors perform zero-shot testing like training on Bench2Drive and testing on nuScenes.
- Training Complexity: The system’s architecture is quite complex, involving multiple stages (VLM pretraining, diffusion planning, RL fine-tuning), which may pose challenges for reproducibility and deployment. Detail training procedures are missing.
- Optional Use of VLM Reasoning: While the VLM can provide high-level reasoning, the paper notes that such guidance did not significantly improve performance in current benchmarks, raising questions about the practical use VLM high-level reasoning. This also raises a question  whether ReCogDrive can be applied to UniAD, VAD like architectures without VLM model, and UniAD-ReCogDrive gain improvements or not on base UniAD performance.

- ORION, ORION: A Holistic End-to-End Autonomous Driving Framework by Vision-Language Instructed Action Generation, ICCV 2025
- VLP, VLP: Vision Language Planning for Autonomous Driving, CVPR 2024.
- DiMA, Distilling Multi-modal Large Language Models for Autonomous Driving, CVPR 2025.

**Questions:**

Please refer weaknesses

---

> ### Author Response · Authors · 2025-11-22
> **Response to Reviewer gJsD 1/3**
>
> We thank the reviewer for the careful evaluation and constructive feedback on our work. We appreciate the recognition of our contributions and address the raised points in detail below.
>
> **1. Comparison with ORION, VLP, and DiMA.**
>
> **Comparison with ORION.** ORION introduces a QT-Former and a VAE-based trajectory decoder, and compares this VAE decoder against a directly transplanted DiffusionDrive module without any further modification or adaptation. To fairly demonstrate the effectiveness of our Cognition-Guided Diffusion Planner, we migrate the VAE Decoder in ORION to our VLM and compare under exactly the same VLM backbone. The results are:
>
> | Method                          |  NC  | DAC  | TTC  | C. |  EP  | PDMS |
> |-|-|-|-|-|-|-|
> | VLM + VAE Decoder              | 97.8 | 91.7 | 93.0 | 100   | 78.0 | 83.0 |
> | VLM + Diffusion Planner (ours) | 98.1 | 94.7 | 94.2 | 100   | 80.9 | 86.5 |
>
> The ORION VAE uses only a single VLM token. Following its design, we feed a learned query embedding into the VLM and use the corresponding output token as the decoder input. Under identical conditioning, the VAE decoder **remains clearly weaker than our diffusion planner**. Beyond performance, it also suffers from two fundamental limitations:
>
> - **Lack of genuine multi-modality.** We evaluate a top-k (k=10) scheme where the model generates multiple candidates and selects the best. The results are:
>
>    | Method                                   |  NC  | DAC  | TTC  | C. |  EP  | PDMS |
>    |-|-|-|-|-|-|-|
>    | VLM + VAE Decoder + Top-k                | 97.8 | 91.8 | 93.0 | 100   | 78.0 | 83.0 |
>    | VLM + Diffusion Planner + Top-k          | 98.7 | 96.3 | 95.8 | 100  | 83.9 | **89.3** |
>
>    For the VAE Decoder, the latent space is constrained to a single Gaussian and heavily regularized toward a unimodal prior, which collapses diverse futures into one dominant mean. The deterministic decoder then maps this narrow latent region to essentially an “average” trajectory, so resampling \(z\) cannot produce genuinely different outcomes. In practice, autonomous driving planners must generate multiple diverse trajectory hypotheses before selecting a safe and comfortable one. This absence of usable multi-modality makes the VAE Decoder unsuitable for such multi-modal planning pipelines.
>
> - **Intractable trajectory likelihood.**  In the ORION-style VAE Decoder, the conditional prior $p(z \mid s_t)$ is itself parameterized by a neural network, and training mainly relies on a KL term between $q(z \mid s_t, y_{1:H})$ and this learned prior. At inference time, however, we do not obtain a tractable, normalized likelihood over trajectories $p(y_{1:H} \mid s_t)$, nor stable log-probabilities for different sampled trajectories. This makes it hard to directly apply GRPO or other policy-gradient style optimization based on trajectory probabilities, especially in safety-critical planning.
>
>    In contrast, our diffusion process provides step-wise denoising objectives and a likelihood surrogate, enabling DiffGRPO and further gains.
>
> **Comparison with VLP and DiMA.** VLP and DiMA distill VLM knowledge into UniAD/VAD-style end-to-end planners, whereas our work follows a VLA formulation where the model directly outputs continuous trajectories. Since neither VLP nor DiMA provides public code, we are unable to benchmark their runtime in our environment. And only DiMA reports latency numbers in its paper. For reference, we **measure ReCogDrive on a single RTX 4090 and obtain 251 ms per sample, which is comparable to DiMA+ (226 ms) and DiMA-Dual+ (286 ms)**. Beyond comparable runtime, our approach offers two additional advantages:
>
> - **Stronger closed-loop performance.**
>    VLP and DiMA both rely on UniAD/VAD as their underlying end-to-end planners. However, these base planners exhibit weak closed-loop driving performance, as shown below:
>
>    | Method            |  DS   |  RC  | PDMS |
>    |-|-|-|-|
>    | UniAD  | 45.8 | 16.4 | 83.4 |
>    | VAD    | 42.35 | 15.0 | 80.9 |
>    | ReCogDrive    | 71.4 | 45.5 | 90.8 |
>
>    Because VLP and DiMA inherit the limitations of their underlying E2E planners, their closed-loop ability is fundamentally constrained. In contrast, ReCogDrive delivers substantially stronger closed-loop driving performance.
>
> - **Plug-and-play reasoning guidance for E2E planners.**
>    Unlike VLP and DiMA, which require jointly training the VLM and the end-to-end model, ReCogDrive allows the VLM’s high-level reasoning to be cached and directly injected into existing planners without retraining the VLM, providing immediate performance gains as shown in Reply 4.
>
> We also report nuScenes open-loop results in Reply 2. Collision rate match DiMA and surpass VLP, though open-loop metrics correlate weakly with real driving quality. Since VLP and DiMA lack official implementations, NAVSIM reproduction is not feasible. We will cite and discuss both methods in the revision and add direct comparisons once official code is released.

---

> ### Author Response · Authors · 2025-11-22
> **Response to Reviewer gJsD 2/3**
>
> **2. Zero-Shot Testing and Real-World Evaluation.** Thank you for this suggestion. First, we would like to clarify that NAVSIM and Bench2Drive together **already provide complementary and realistic evaluation settings for our method**. NAVSIM uses **real-world driving logs** combined with a non-reactive simulator to evaluate sensor-based policies with simulation-based safety metrics such as collisions and map compliance, effectively serving as a real-world–grounded closed-loop benchmark. In contrast, Bench2Drive is a **closed-loop CARLA-based benchmark** that provides a large-scale, quasi-realistic simulation environment with diverse interactive scenarios, weathers, and towns for end-to-end autonomous driving evaluation.
>
> Second, regarding the proposed zero-shot setting “training on Bench2Drive and testing on nuScenes”, we believe this is not a scientifically fair protocol: Bench2Drive is a CARLA-based simulation while nuScenes is collected in the real world, and the two datasets exhibit significantly different vehicle dynamics and behavior distributions. Even for nominally similar scenarios, the underlying driving style (e.g., speeds, accelerations, braking patterns) differs substantially. To illustrate this mismatch, we compare some basic kinematic statistics below:
>
> | Dataset      | Mean Speed (m/s)  | Mean acc (m/s²) | Mean yaw_rate (rad/s) | Low-speed Fraction |
> |:-|:-:|:-:|:-:|:-:|
> | Bench2Drive |       3.13        |        0.75        |          0.32           |        0.42         |
> | NAVSIM      |       4.44        |        0.67        |          0.19           |        0.11         |
> | nuScenes    |       3.40       |        0.50        |          0.37           |        0.31         |
>
> These statistics reveal substantial gaps in kinematics, **making Bench2Drive→nuScenes zero-shot results reflect dataset shift rather than generalization**. Moreover, camera intrinsics/extrinsics, sensor configurations, and ego-vehicle parameters also differ across the datasets, further compounding the mismatch.
>
> To still provide insight into cross-dataset behavior, we include a supplementary experiment where ReCogDrive is pre-trained on Bench2Drive and evaluated on nuScenes in both zero-shot and fine-tuned settings:
>
>
> | Method                               | L2@1s | L2@2s | L2@3s | Avg. L2 | Coll.@1s | Coll.@2s | Coll.@3s | Avg. Coll. |
> |:-|:-:|:-:|:-:|:-:|:-:|:-:|:-:|:-:|
> | UniAD                  | 0.20  | 0.42  | 0.75  | 0.46    | 0.02     | 0.25     | 0.84     | 0.37       |
> | VLP                    | 0.30  | 0.53  | 0.84  | 0.55    | 0.01     | 0.07     | 0.38     | 0.15       |
> | VAD                    | 0.17  | 0.34  | 0.60  | 0.37    | 0.07     | 0.10     | 0.24     | 0.14       |
> | DriveVLM               | 0.18  | 0.34  | 0.68  | 0.40    | 0.10     | 0.22     | 0.45     | 0.27       |
> | DiMA-Dual+             | 0.14  | 0.27  | 0.46  | 0.29    | 0.05     | 0.07     | 0.15     | 0.09       |
> | AutoVLA                | 0.22 | 0.39  | 0.61  | 0.41    | 0.10     | 0.17     | 0.28     | 0.18       |
> | ReCogDrive (zero-shot) | 1.34  | 1.91  | 2.69  | 1.98    | 0.07     | 0.57     | 0.82     | 0.82       |
> | ReCogDrive (fine-tune) | 0.17  | 0.34  | 0.63  | 0.38    | 0.07     | 0.09     | 0.16     | 0.10       |
>
> Note that all other methods are trained/validated on nuScenes, while our “zero-shot” result is out-of-domain (Bench2Drive→nuScenes); the fine-tuned result is the comparable setting. From the table, zero-shot evaluation **suffers a large performance drop due to the severe cross-domain mismatch**, while fine-tuning on nuScenes substantially **closes this gap and restores competitive performance**. In particular, the **average collision rate drops to 0.10**, which is **comparable to other SOTA end-to-end planners**, demonstrating the strong generalization capability of our model.
>
> In addition, as noted in prior work (such as AD-MLP and BEV-Planner), nuScenes open-loop L2 metrics provide only a weak indication of real driving capability. For this reason, we place greater emphasis on the closed-loop metrics reported on NAVSIM and the closed-loop performance on Bench2Drive, which offer a more reliable evaluation of practical driving quality.
>
>
>
> **3. Training Complexity and Reproducibility.** Thank you for raising this point. We agree that clarity in training procedures is crucial. In the supplementary material (Experiment Details), we already provide detailed descriptions of the model architecture, hyperparameters, and training configurations for all stages, including VLM pretraining, diffusion planning, and RL fine-tuning.
>
> Following your suggestion, we **have further expanded this section and added more implementation details** in the updated paper to ensure full reproducibility. We also guarantee that **all datasets, code, and trained models used in this work will be fully released** to support reliable replication and future research.

---

> ### Author Response · Authors · 2025-11-22
> **Response to Reviewer gJsD 3/3**
>
> **4. VLM Reasoning and Applicability to UniAD/VAD-Style Architectures.** We thank the reviewer for raising this question about the practical role of VLM reasoning and the applicability of ReCogDrive to end-to-end architectures. On NAVSIM, the limited additional gains mainly reflect the benchmark itself and the fact that, after DiffGRPO fine-tuning, the diffusion planner already produces stable and safe trajectories, rather than a lack of value in VLM high-level reasoning.
>
> Regarding whether ReCogDrive can be applied to end to end models such as UniAD or VAD, our answer is yes. Since there are no publicly available UniAD or VAD implementations on NAVSIM, we combine ReCogDrive with two other E2E planners, Transfuser and DiffusionDrive, to obtain Transfuser-ReCogDrive and DiffusionDrive-ReCogDrive. In these variants, the ReCogDrive-VLM outputs high-level driving intents (for example, accelerate, decelerate, change lane left or right) and these intents are encoded as meta action embeddings that are injected into the end-to-end planner to guide trajectory generation.
>
> The results are:
>
> | Method                      |  NC  | DAC  | TTC  | C. |  EP  | PDMS |
> |:-:|:-:|:-:|:-:|:-:|:-:|:-:|
> | Transfuser                  | 97.7 | 92.8 | 92.8 | 100   | 79.2 | 84.0 |
> | Transfuser-ReCogDrive       | 98.0 | 95.1 | 94.2 | 100   | 81.2 | 86.7 |
> | DiffusionDrive              | 98.2 | 96.2 | 94.8 | 100   | 82.2 | 88.1 |
> | DiffusionDrive-ReCogDrive   | 98.9 | 96.4 | 95.9 | 100   | 83.0 | 89.0 |
>
> **These gains demonstrate that ReCogDrive-VLM high-level reasoning can effectively enhance end-to-end planners, highlighting its practical value for improving autonomous driving safety and overall performance.**

---

### Official Review · Reviewer_RX3X · 2025-11-03

**Soundness:** 3
**Presentation:** 3
**Contribution:** 2
**Rating:** 6
**Confidence:** 4

**Summary:**

This paper proposes ReCogDrive, an end-to-end autonomous driving framework that combines a Vision-Language Model (VLM) with a diffusion-based planner and an RL fine-tuning stage (DiffGRPO). The key idea is to use a cognitive data pipeline to teach the VLM human-like reasoning skills, then guide continuous trajectory generation through diffusion and reinforcement learning. The experiments on NAVSIM and Bench2Drive look solid — the model consistently outperforms both VLM- and diffusion-based baselines and achieves strong closed-loop results.

**Strengths:**

The paper is well-written and easy to follow, with clear figures and good motivation.
I like the overall integration of cognitive reasoning (via VLM) and low-level control (via diffusion + RL).
Results are strong and consistent across benchmarks, and the ablation studies help justify each component.
Technically, the approach is sound.

**Weaknesses:**

**(a) Novelty Overlap / Incremental Concerns**

Several very recent works (e.g., Drive-R1 Li et al., 2025 and AlphaDrive Jiang et al., 2025) already explore reinforcement learning and reasoning within VLM-based driving.

Likewise, Gen-Drive (Huang et al., 2025) combines diffusion with RL for driving policy optimization.

Given these, the claim of “first to apply reinforcement learning to VLA models” may be over-stated, and the conceptual contribution, though strong in integration, is not entirely novel.

**(b) Ablation Limitations**

No comparison against other RL-enhanced planners such as under identical environments, limiting attribution of improvements to DiffGRPO specifically.

The reward design (PDMS) could be discussed in greater depth—its weighting and sensitivity are unspecified.

**Questions:**

1. Could the authors provide the weighting coefficients used for the PDMS reward and justify their selection?

2. Is the VLM backbone frozen during diffusion and RL training, or does it receive gradient updates through the planner?

3. What is the exact runtime efficiency (latency per frame) compared to baseline diffusion and text-based methods?

4. Can the authors show qualitative examples of failure modes or collisions to highlight what DiffGRPO specifically improves?

---

> ### Author Response · Authors · 2025-11-22
> **Response to Reviewer RX3X 1/3**
>
> We thank the reviewer for the valuable suggestions. We have provided the requested comparisons with relevant methods and clarified the corresponding technical details.
>
> **1. Comparison with Recent RL-Based Driving Works.** We appreciate the reviewer’s feedback. First, **we would like to clarify that AlphaDrive applies RL to VLMs rather than VLAs, Drive-R1 is a later work that appeared after our submission, and Gen-Drive applies RL only to an end-to-end driving model**. We acknowledge that the phrasing “first to apply reinforcement learning to VLA models” can be misleading, and **we have removed it from both the main paper and the appendix.**
>
> **Regarding the Drive-R1 and AlphaDrive works you mentioned**, our contribution is to address two persistent issues in VLM-based driving methods that output text-formatted trajectories or high-level commands (e.g., EMMA, Drive-R1, AlphaDrive): (i) the generated textual trajectories may violate the required templates, leading to parsing failures that are unacceptable even at low frequency, and (ii) slow reasoning latency (see Tab. 4 in the main paper). ReCogDrive couples a VLM with a cognition-guided diffusion planner to generate stable, safe continuous trajectories efficiently, and then applies DiffGRPO to further optimize safety and comfort via interaction with the environment, while the VLM can optionally output high-level commands or CoT reasoning to guide the planner. This differs from Drive-R1 and AlphaDrive, which primarily use RL to improve chain-of-thought reasoning, whereas our method couples a cognition-guided diffusion planner with a tailored RL algorithm (DiffGRPO) to jointly optimize the continuous planning policy for safety and comfort. To make this more concrete, we also compare a Drive-R1-like baseline and AlphaDrive on NAVSIM:
>
> | Method                             |  NC  | DAC  | TTC  | C. |  EP  | PDMS | Inference Time (s/frame) |
> |:-|:-:|:-:|:-:|:-:|:-:|:-:|:-:|
> | InternVL + GRPO + CoT (Drive-R1-like baseline) | 98.2 | 96.4 | 94.6 | 100   | 84.2 | 88.9 | 0.86                      |
> | AlphaDrive + DiffusionDrive | 98.3 | 97.6 | 95.4 | 100   | 83.1 | 89.5 | N/A |
> | ReCogDrive                         | 97.9 | 97.3 | 94.9 | 100   | 87.3 | **90.8** | **0.08**                      |
>
> Although Drive-R1 did not report results on NAVSIM, we implement a Drive-R1-like baseline following its design for a fair comparison. For AlphaDrive, the PDMS score is quoted from the paper, which does not provide inference-time measurements. As shown, **ReCogDrive achieves substantially lower latency together with a higher PDMS, demonstrating the effectiveness of our approach.**
>
> **Compared with Gen-Drive**, our focus is specifically on the VLA setting and on the above failure modes of current VLM-driven approaches. Our contribution lies in building a scalable hierarchical data pipeline to inject driving cognition into the VLM, coupling this cognitively trained VLM with a diffusion planner for efficient trajectory planning, and introducing DiffGRPO to further refine safety and comfort.
>
> **We already cite Gen-Drive and AlphaDrive in Related Work and have added Drive-R1 for completeness.** We appreciate the reviewer bringing this to our attention.
>
> **2.1. Comparison with Other RL-Enhanced Planners.** We thank the reviewer for the helpful comments. To better verify the effectiveness of DiffGRPO, we compare ReCogDrive with several RL-enhanced planners under the same NAVSIM environment. **Notably, many of these methods were developed after or concurrently with our work.** The results are as follows:
>
> | Method                      | Ref            |  NC  | DAC  | TTC  | C. |  EP  | PDMS |
> |:-|:-:|:-:|:-:|:-:|:-:|:-:|:-:|
> | Trajhf(DPGRPO)              | arXiv 2025     | 96.6 | 96.6 | 92.1 | 100   | 84.5 | 87.6 |
> | CoIRL-AD                    | arXiv 2025     | 98.6 | 96.8 | 95.5 | 100   | 81.0 | 88.2 |
> | DIVER(GRPO)                 | arXiv 2025     | 98.5 | 96.5 | 94.9 | 100   | 82.6 | 88.3 |
> | AutoVLA(3B)(GRPO)           | NeurIPS 2025   | 98.4 | 95.6 | 98.0 | 99.9  | 81.9 | 89.1 |
> | AlphaDrive(8B)(GRPO)        | arXiv 2025     | 98.3 | 97.6 | 95.4 | 100   | 83.1 | 89.5 |
> | DriveDPO                    | NeurIPS 2025   | 98.5 | 98.1 | 94.8 | 99.9  | 84.3 | 90.0 |
> | InternVL(8B) + GRPO         | —              | 98.2 | 96.1 | 94.3 | 100   | 83.5 | 88.3 |
> | InternVL(8B) + Think + GRPO | —              | 98.2 | 96.4 | 94.6 | 100   | 84.2 | 88.9 |
> | ReCogDrive(2B)              | —              | 97.9 | 97.3 | 94.9 | 100   | 87.3 | **90.8** |
>
> As can be seen, ReCogDrive not only outperforms VLA-based RL methods such as AutoVLA (3B, GRPO-based fast-slow thinking), AlphaDrive (8B, GRPO) with DiffusionDrive, and the InternVL (8B) + GRPO baseline, but also achieves a higher PDMS than other end-to-end RL-enhanced planners such as TrajHF (DPGRPO) and DriveDPO (DPO), **further supporting the effectiveness of DiffGRPO under identical evaluation conditions.**

---

> ### Author Response · Authors · 2025-11-22
> **Response to Reviewer RX3X 2/3**
>
> **2.2. Regarding the Reward Design.** We appreciate the reviewer’s questions regarding reward design. Our PDMS reward is constructed from five components: EP, TTC, Comfort (C), NC, and DAC. Concretely, the PDMS is computed as:
>
> $$
> \text{PDMS} = \text{NC} \times \text{DAC} \times
> \left(
> \frac{ w_{\text{EP}} \cdot \text{EP} + w_{\text{TTC}} \cdot \text{TTC} + w_{\text{C}} \cdot \text{C} }
> { w_{\text{EP}} + w_{\text{TTC}} + w_{\text{C}} }
> \right)
> $$
>
> where $w_{\text{EP}}, w_{\text{TTC}}, w_{\text{C}}$ are the weighting coefficients for EP, TTC, and Comfort, respectively, and in NAVSIM the official default setting assigns them as $w_{\text{EP}} = 5.0$, $w_{\text{TTC}} = 5.0$, and $w_{\text{C}} = 2.0$. NC, DAC, EP, TTC, and C together measure whether the ego vehicle avoids at-fault collisions (NC), stays within drivable areas (DAC), makes sufficient progress along the planned route (EP), keeps a safe time gap to others (TTC), and maintains human-like comfort in acceleration/jerk (C). After imitation learning, the model already achieves very few collisions and maintains comfortable driving behavior, resulting in consistently high NC and Comfort scores. Therefore, we primarily study how the weights of **EP ($w_{\text{EP}}$)** and **TTC ($w_{\text{TTC}}$)** influence the model’s performance, as shown in the table below.
>
>
> | $w_{\text{EP}}$ | $w_{\text{TTC}}$ | NC | DAC | TTC | C. | EP | PDMS |
> |:-:|:-:|:-:|:-:|:-:|:-:|:-:|:-:|
> | 5.0  | 5.0  | 98.6 | 97.8 | 96.2 | 100 | 84.0 | 90.3 |
> | 10.0 | 5.0  | 97.9 | 97.3 | 94.9 | 100  | 87.3 | 90.8 |
> | 20.0 | 5.0  | 97.8 | 97.3 | 94.2 | 100 | 87.3 | 90.5 |
> | 30.0 | 5.0  | 97.6 | 97.2 | 93.1 | 99.6 | 87.6 | 90.0 |
> |      |      |      |      |      |      |      |      |
> | 10.0 | 5.0  | 97.9 | 97.3 | 94.9 | 100  | 87.3 | 90.8 |
> | 10.0 | 10.0 | 98.5 | 97.7 | 95.9 | 100 | 85.1 | 90.5 |
> | 10.0 | 20.0 | 98.6 | 97.7 | 96.2 | 100 | 84.5 | 90.5 |
> | 10.0 | 30.0 | 98.5 | 97.8 | 96.1 | 100 | 84.5 | 90.4 |
>
> As shown in the table, increasing the EP weight $w_{\text{EP}}$ makes the policy more aggressive: EP increases, but NC/DAC/TTC slightly drop, leading to reduced overall PDMS when $w_{\text{EP}}$ is too large. Conversely, increasing the TTC weight $w_{\text{TTC}}$ yields higher TTC (more conservative, safer gaps) but lowers EP, again saturating PDMS. This ablation explicitly characterizes the weighting and sensitivity of PDMS under a wide range of $w_{\text{EP}}$ and $w_{\text{TTC}}$ settings. Based on this analysis, we therefore choose **$w_{\text{EP}} = 10.0$** and **$w_{\text{TTC}} = 5.0$** as our default coefficients, which provide the best trade-off between progress and safety and yield the highest PDMS. We now explicitly report both this configuration and the above sensitivity analysis in the supplementary material.
>
>
> **3. PDMS Reward Weighting Coefficients.** As discussed in **Reply 2.2**, we have provided a detailed explanation of the PDMS weighting coefficients and their effects. In brief, our default setting uses **$w_{\text{EP}} = 10.0$**, **$w_{\text{TTC}} = 5.0$**, and **$w_{\text{C}} = 2.0$**, which we found to offer the best trade-off between progress and safety. **The full formulation, coefficient settings, and sensitivity analysis are now included in the supplementary material and are clearly highlighted in the revised version.**
>
>
>
> **4. Whether the VLM Backbone is Trained in the Diffusion and RL Stages.** Appreciate the reviewer’s questions on these details. During both the diffusion-based imitation learning (IL) and reinforcement learning (RL) stages, we keep the VLM backbone frozen, meaning it does not receive gradient updates through the planner. **This setting is explicitly stated in the Implementation Details of the main paper.**
>
> We adopt this design for two main reasons.
> (1) Updating the VLM backbone during diffusion learning harms its reasoning ability. Using diffusion-generated trajectories as training targets can cause the autoregressively trained VLM to lose its language capability. To preserve the VLM’s driving priors and reasoning skills, we keep the backbone frozen during both IL and RL.
>
> (2) Freezing the backbone improves planning performance and greatly accelerates training. We further compare training with and without freezing the VLM backbone during diffusion imitation learning. As shown below, freezing the backbone achieves a higher PDMS and allows caching the VLM features, reducing training time.
>
> | VLM Backbone | Diffusion Planner | Cache Features | Training Time      | PDMS |
> |:-:|:-:|:-:|:-:|:-:|
> | Training     | Training          | No              | 1.0× (baseline)    | 74.8        |
> | Freeze       | Training          | Yes             | 0.04×           | **86.5**    |
>
> These results confirm that freezing the VLM backbone not only preserves reasoning ability but also improves efficiency and overall planning performance.

---

> ### Author Response · Authors · 2025-11-22
> **Response to Reviewer RX3X 3/3**
>
> **5. Runtime Efficiency Comparison.** Thank you for the question. We apologize for the lack of clarity. In our paper, the “Time (s)” column in Tab. 4 already corresponds to the **latency per frame**, i.e., the wall-clock inference time for processing **one sample (one frame)** and predicting a full trajectory.
>
> For completeness, we now explicitly clarify our runtime measurement protocol. All timings are measured on an NVIDIA H200 GPU with batch size 1, where the input is a single front-view camera frame and the output is one predicted trajectory. Our diffusion planner uses 5 DDIM steps, and the reported number is the end-to-end latency per frame. Under this setup, **our method achieves 0.08 s per frame**, which corresponds to the entries shown in Tab. 4. For comparison, text-based trajectory prediction via VLM decoding is much slower (**0.58 s per frame**), while **the baseline diffusion planner runs at a similar speed** to our improved diffusion planner but with lower PDMS. We have updated the paper to explicitly define the latency measurement protocol and to include these implementation details, in order to avoid ambiguity.
>
> **6. Qualitative Failure Case Analysis.** Thank you for prompting qualitative analysis. **We have added extensive qualitative analyses in Fig. 5 (main paper) and Figs. 10–14 (appendix).** As shown in the figures, methods trained purely with imitation learning such as Transfuser and ReCogDrive-IL frequently exhibit failure cases including insufficient turning angles that lead to curb collisions, delayed turning, incorrect lane choices at intersections, and difficulty maintaining consistent lane adherence. In contrast, DiffGRPO produces trajectories that consistently avoid these issues, demonstrating safer turning behavior, more reliable lane placement, and overall more stable and collision-free planning.

---

### Author Response · Authors · 2025-12-02
**Rebuttal Summary for Area Chair**

Dear Area Chair,

We sincerely thank you and all reviewers for the time and effort spent evaluating our submission and for the constructive feedback.

We also appreciate the positive assessment of our work, including the clear motivation and presentation, the integration of a cognitive VLM with a diffusion-based continuous planner and an RL refinement stage, and the strong empirical results (with comprehensive ablations) on challenging benchmarks such as NAVSIM and Bench2Drive. Below, we briefly summarize how we addressed the main concerns raised in the initial reviews.

**Reviewer RX3X** mainly raised concerns about overlap with recent driving RL works, missing comparisons with other RL-enhanced planners, reward and training details, runtime, and the need for qualitative analysis. We addressed these by highlighting key distinctions from recent driving RL works and empirically validating the performance gains **(R1)**, adding comparisons with other RL-enhanced planners **(R2.1)**, providing the PDMS reward weighting and sensitivity analysis **(R2.2 and R3)**, explicitly stating the frozen-VLM setup **(R4)**, clarifying the latency definition in Tab. 4 **(R5)**, and adding qualitative analyses to illustrate the benefits of DiffGRPO **(R6)**.

**Reviewer gJsD** mainly raised concerns about missing comparisons with ORION, VLP, and DiMA, zero-shot evaluation, reproducibility, and the practical value of VLM reasoning for E2E planners. We addressed these by adding a fair comparison with ORION and discussing VLP/DiMA with additional runtime context **(R1)**, clarifying the roles of NAVSIM and Bench2Drive and adding a cross-dataset evaluation on nuScenes while explaining why the zero-shot setting is unfair **(R2)**, expanding implementation details to improve reproducibility **(R3)**, and demonstrating plug-and-play VLM intent guidance that improves other E2E planners **(R4)**.

**Reviewer diW3** mainly raised concerns about DiffGRPO justification, clarity of the method formulation, and missing details/ablations. We addressed these by motivating GRPO for multi-modal driving and adding comparisons with REINFORCE and DPPO under the same setup **(R1)**, clarifying DiffGRPO via an MDP formulation and explaining the group construction with shared initial conditioning **(R2)**, expanding the data pipeline and diffusion planner details with examples and implementation specifics **(R3)**, fixing notation and defining previously ambiguous symbols **(R4)**, precisely defining the inputs/outputs and explaining why headings are needed **(R5)**, and justifying the roles of token-level conditioning and pooled global embedding with ablations **(R6)**.

**Reviewer wtRP** mainly raised concerns about novelty, the role of cognition, fairness of PDMS-based RL, missing VLA baselines, and protocol clarity. We addressed these by demonstrating the novelty and effectiveness of our data pipeline, diffusion planner, and DiffGRPO relative to prior methods **(R1)**, demonstrating the effectiveness of driving pre-training, showing the cognitive role of the RL stage, and verifying that VLM reasoning improves E2E planners **(R2)**, explaining the rationale and common use of PDMS as a reward and validating effectiveness on Bench2Drive **(R3)**, adding direct comparisons with AutoVLA and reporting latency **(R4)**, and specifying the training data composition and clarifying protocol definitions for clarity **(R5)**.

Overall, these revisions clarify our positioning and methodology, strengthen the empirical evidence and comparisons, and improve transparency and reproducibility. We believe the revised manuscript is more complete and comprehensive, and better supports the paper’s contributions and conclusions.

Best regards,

The Authors of Submission 1237

---

### Meta-Review · Area_Chair_2Cpq · 2026-01-07

**Summary:**

This paper proposes ReCogDrive, a framework that integrates Vision Language Models (VLMs) with a diffusion planner and reinforcement learning to address the language-action mismatch in autonomous driving. The method achieves strong empirical performance on NAVSIM. The core debate during the review process revolved around the novelty of the framework relative to concurrent works and the justification for the specific reinforcement learning choices.

**Reviewer Concerns:**

The main concerns were comparisons with recent RL works (RX3X), justification of the optimization algorithm (diW3), and the distinct role of the VLM (wtRP). The rebuttal substantively addresses these by adding baselines against REINFORCE and DPPO (diW3), providing runtime analysis (RX3X), and comparing with ORION (gJsD). While wtRP questioned the novelty and VLM role, the new comparisons with AutoVLA and detailed ablation studies demonstrate empirical value that arguably outweighs the framing concerns.

**Reviewer Scores:**

Given the comprehensive rebuttal, I expect RX3X and diW3 to raise their scores to accept due to the clarified positioning and theoretical motivation. gJsD will likely maintain their acceptance. Although wtRP did not participate in the post-rebuttal discussion, the added AutoVLA comparisons and ablations effectively address their original concerns, rendering the initial low score of 4 outdated.

---

### Decision · Program_Chairs · 2026-01-26

Accept (Poster)